# Extending Prediction-Powered Inference through Conformal Prediction

## Abstract

Prediction-powered inference is a recent methodology for the safe use of black-box ML models to impute missing data, strengthening inference of statistical parameters. However, many applications require strong properties besides valid inference, such as privacy, robustness or validity under continuous distribution shifts; deriving prediction-powered methods with such guarantees is generally an arduous process, and has to be done case by case. In this paper, we resolve this issue by connecting prediction-powered inference with conformal prediction: by performing imputation through a calibrated conformal set-predictor, we attain validity while achieving additional guarantees in a natural manner. We instantiate our procedure for the inference of means, Z- and M-estimation, as well as e-values and e-value-based procedures. Furthermore, in the case of e-values, ours is the first general prediction-powered procedure that operates off-line. We demonstrate these advantages by applying our method on private and time-series data. Both tasks are nontrivial within the standard prediction-powered framework but become natural under our method.

## 1 Introduction

Quality statistical inference requires a considerable amount of samples, which can be difficult to obtain or may be missing. Prediction-powered inference (Angelopoulos et al., 2023a) is a recent and promising approach that addresses this challenge by using a black-box ML model to predict the missing samples from the auxiliary data, while simultaneously correcting for the bias induced by this imputation. However, many practical applications require the resulting inferences to satisfy strong guarantees beyond validity, such as privacy (for sensitive data), robustness (to protect against outliers or distribution shifts) or validity under continuously changing scenarios. Deriving prediction-powered methods that satisfy requirements of this sort remains challenging, with existing work relying on case-by-case constructions.

In this paper, we resolve this by connecting prediction-powered inference with conformal prediction. In particular, we show that a calibrated set-predictor can be used for prediction-powered inference in a general manner, while inheriting additional properties from a conformal calibration procedure; this allows us to directly leverage the vast literature on conformal prediction with additional guarantees, spanning privacy (Angelopoulos et al., 2021; Penso et al., 2025), robustness to strategic and adversarial distribution shift (Csillag et al., 2024; Zargarbashi & Bojchevski, 2025; Massena et al., 2025), continuous distribution shift (Gibbs & Candès, 2021; Zaffran et al., 2022; Angelopoulos et al., 2024; Areces et al., 2025), robustness to outliers (Clarkson et al., 2024; Peng et al., 2025; Feldman et al., 2025), censored/missing data (Zaffran et al., 2023; Davidov et al., 2025) and many more. In this way, we offer a single, general solution that overcomes the fragmented, case-specific nature of previous works.

We develop our approach for the inference of means, Z- and M-estimation problems, as well as e-values and e-value-based procedures. This is the first general method for prediction-powered inference with additional guarantees, as well as the first instance of conformal prediction being used for nonparametric statistical inference. When existing prediction-powered methods are applicable, their performance is close to ours. We illustrate our approach in two settings beyond the scope of previous methods, highlighting its advantages.

**Our contributions**

- We propose a general framework for deriving prediction-powered methods with stronger guarantees such as privacy, robustness and validity under continuous distribution shift. Our method works by performing imputation through a calibrated conformal set-predictor; these guarantees are then directly achieved by choosing an appropriate conformal calibration method, for which a substantial body of work exists. Our framework's ability to inherit properties from conformal prediction methods renders it immediately applicable in many diverse settings, where previous prediction-powered methods fall short.

- We instantiate our framework for (i) inference of means; (ii) general Z- and M-estimation problems; and (iii) general e-values and e-value-based procedures – thus matching the breadth of existing prediction-powered methods. In each case, we prove that our procedure is valid under minimal assumptions and quantify their statistical power, which we find to be directly linked to the average size of the conformal predictive sets and their miscoverage rate. Furthermore, in the setting of e-values, our procedure is the first general prediction-powered inference procedure valid without active data collection.

- Beyond comparisons with existing prediction-powered methods, we apply our approach to two practical settings out of reach of prior work: (i) private healthcare for thyroid cancer, and (ii) continuous risk monitoring of a deployed model. In each setting we obtain procedures that can be readily applied by practitioners. In both accounts, ours is the first applicable prediction-powered procedure, thus setting an important baseline for future work.

## 1.1 RELATED WORK

**Prediction-powered inference**  In many applications, researchers have access to large datasets but only small amounts of expensive ground truth 'labels.' Though machine learning models can often accurately predict labels for the whole dataset, they are not perfect; in particular, statistical inference atop such predictions can suffer from significant bias. Prediction-powered inference seeks to resolve this, by appropriately debiasing such inferences. The topic already spans a significant body of work both methodological (e.g., (Angelopoulos et al., 2023a;b; Fisch et al., 2024; Gu & Xia, 2024; Ji et al., 2025; Csillag et al., 2025; Cortinovis & Caron, 2025)) and applied (e.g., (Boyeau et al., 2024; Aiken et al., 2025)). Existing methods typically prove valid inference (i.e., lack of bias), with some works also establishing guarantees under covariate or label shift. Towards additional guarantees (e.g. privacy, robustness, etc.), the works of (Li et al., 2025; Luo et al., 2024; Hays & Raghavan, 2025) establish guarantees under performativity, federation and interference, respectively, but require ad-hoc analyses to do so.

**Conformal prediction**  On the other side of the literature, conformal prediction (Vovk et al., 2005) has emerged as a solid manner of quantifying uncertainty about predictions. In its most common formulation, conformal prediction produces for each sample a predictive set that will contain the true label with probability at least $1 - \alpha$, for a significance level $\alpha \in (0, 1)$ chosen a priori. Conformal prediction has also spanned a vast amount of literature on methodology (e.g., (Tibshirani et al., 2019; Angelopoulos et al., 2022; Gibbs et al., 2023; Csillag et al., 2024; van der Laan & Alaa, 2024)), theory (e.g., (Kiyani et al., 2025; Bian & Barber, 2022)) and applications (e.g., (Zhou et al., 2022; Csillag et al., 2023; Genari & Goedert, 2025)); of particular relevance is the wide literature on conformal prediction with additional guarantees, e.g. (Angelopoulos et al., 2021; Penso et al., 2025; Csillag et al., 2024; Zargarbashi & Bojchevski, 2025; Massena et al., 2025; Gibbs & Candès, 2021; Zaffran et al., 2022; Angelopoulos et al., 2024; Areces et al., 2025; Clarkson et al., 2024; Peng et al., 2025; Feldman et al., 2025; Zaffran et al., 2023; Davidov et al., 2025).

**Connecting the two**  Though the two tackle similar problems, connecting them is not immediate: conformal prediction guarantees that the probability of a single predictive set containing its corresponding label is high, but statistical inference requires multiple data points, not a single one. A back-of-the-envelope calculation would give us that, if conformal prediction ensures that a single prediction set will contain its corresponding true value with probability $1 - \alpha$, the probability that $n$ independent prediction sets will contain their corresponding true values will be of about $(1 - \alpha)^n$, which quickly becomes problematic as $n$ grows. Indeed, various works have tried to alleviate this issue for "batch" conformal prediction (Gazin et al., 2024; Guille-Escuret & Ndiaye,

2024; Jin & Candès, 2022; Marandon, 2023), sometimes even with the explicit goal of statistical inference (Guille-Escuret & Ndiaye, 2024). However, they all reach an overarching conclusion that one would need to adjust the conformal predictor in a manner that still scales badly as $n$ grows. Our work, in contrast, requires no such adjustment.

**Conformal prediction for statistical inference** Conformal prediction has spanned much work, but relatively little in regards to its connections to more usual statistical inference. Of particular note is (Guille-Escuret & Ndiaye, 2024), which leverages conformal prediction for inference of the parameter $\theta$ of a statistical model of the form $Y = f_\theta(X) + \xi$ through a voting mechanism. However, besides being limited to this specific statistical model and task, it requires harsh assumptions on the nature of the noise $\xi$ that make it sensitive to misspecification. Also worth highlighting is the work of (Cabezas et al., 2024), which uses ideas from conformal prediction to solve statistical inference problems, but is not applicable to prediction-powered inference.

## 2 METHOD

We first present our method in the simple context of mean estimation. Then, building up on the idea of conformal prediction-powered mean estimation we extend to progressively more complex settings, first considering Z- and M-estimation tasks (e.g., means, quantiles and regression coefficients), and then general e-value-powered procedures.

Throughout this section, we consider that we have i.i.d. data $(X_i, Y_i)_{i=1}^n \sim P$ from some unknown distribution $P$, where we have access to the $X_i$ but the $Y_i$ are missing. Leveraging the i.i.d. assumption, we will additionally make use of $(X, Y) \sim P$ when the indices are irrelevant, and denote the support of these variables by $\mathcal{X} = \text{supp}(X)$ and $\mathcal{Y} = \text{supp}(Y)$. In particular sections some additional assumptions are necessary, and will be made accordingly.

Let $C : \mathcal{X} \to 2^{\mathcal{Y}}$ be a set-predictor, fit on some hold-out data; we define its miscoverage rate $\text{Err}(C) := \mathbb{P}[Y \notin C(X)]$. It is known that when $C$ is fit via, e.g., split conformal prediction with target miscoverage $\gamma \in (0, 1)$, we will have $\text{Err}(C) \approx \gamma$ (Angelopoulos & Bates, 2021; Bian & Barber, 2022). For full generality, we consider the conformal predictor fixed and state our results in terms of just $\text{Err}(C)$.

For the sake of clarity, we keep our presentation in the main paper purely to scalar estimation problems. Multivariate estimation follows analogously; see Appendix B.2.

### 2.1 WARMUP: MEAN ESTIMATION

Our goal here is to infer $\mathbb{E}[\phi(Y)]$ for some function $\phi : \mathcal{Y} \to \mathbb{R}$; for this we will need to assume that $\phi(Y)$ is bounded almost surely within some interval $[a, b]$. For convenience, let $\phi(C(X)) := \{\phi(y) : y \in C(X)\}$ and $M = b - a$. It then follows:

**Lemma 2.1.** *Let $C : \mathcal{X} \to 2^{\mathcal{Y}}$ be a set predictor and suppose that $\phi(Y) \in [a, b]$ almost surely. Then*

$$\mathbb{E}[\inf \phi(C(X))] - M \, \text{Err}(C) \leq \mathbb{E}[\phi(Y)] \leq \mathbb{E}[\sup \phi(C(X))] + M \, \text{Err}(C).$$

*Proof sketch.* We will show that $\mathbb{E}[\inf \phi(C(X))] - M \, \text{Err}(C) \leq \mathbb{E}[\phi(Y)]$ by showing that $\mathbb{E}[\inf \phi(C(X)) - \phi(Y)] \leq M \, \text{Err}(C)$. The proof of the upper bound is analogous, and can be found in the appendix.

The key idea is to use the law of total expectation to condition on whether $Y$ belongs in the predictive set $C(X)$:

$$\mathbb{E}[\inf \phi(C(X)) - \phi(Y)] = \mathbb{E}[\inf \phi(C(X)) - \phi(Y) | Y \in C(X)] \, \mathbb{P}[Y \in C(X)]$$
$$+ \mathbb{E}[\inf \phi(C(X)) - \phi(Y) | Y \notin C(X)] \, \mathbb{P}[Y \notin C(X)];$$

Now, given that $Y \in C(X)$, it must hold that $\phi(Y) \in \phi(C(X))$, and so $\inf \phi(C(X)) \leq \phi(Y)$; thus $\mathbb{E}[\inf \phi(C(X)) - \phi(Y) | Y \in C(X)] \leq 0$. Additionally, note that because both $\phi(C(X))$ and $\phi(Y)$ are bounded in $[a, b]$, it holds that $\inf \phi(C(X)) - \phi(Y) \leq b - a = M$ almost surely, and so $\mathbb{E}[\inf \phi(C(X)) - \phi(Y) | Y \notin C(X)] \leq M$. Thus

$$\mathbb{E}[\inf \phi(C(X)) - \phi(Y)] \leq 0 + M \, \text{Err}(C) = M \, \text{Err}(C). \qquad \square$$

*Remark* 2.2. The assumption that the image of $\phi$ be bounded seems necessary. If it is not, then for $\mathbb{E}[\inf \phi(C(X)) - \phi(Y)|Y \notin C(X)]$ to be well-behaved will generally require relatively strong assumptions on the underlying predictive model and data distribution. That said, it is still possible to infer unbounded means with our framework, just not with this method: see Appendix B.6 for how e-values enable this.

Note that $\mathrm{Err}(C)$ is controlled by the conformal calibration, and that it is independent from the size $n$ of the data set for inference, and thus this bound scales gracefully.

Lemma 2.1 establishes that the means can be safely bounded via imputations based on our conformal predictive sets. This motivates the following procedure:

(i) Fit the conformal set predictor $C$ on a hold-out dataset with some conformal calibration method (e.g. split conformal prediction);

(ii) Use the unlabelled data $(X_i)_{i=1}^n$ to compute lower and upper one-sided $(1 - \alpha/2)$-confidence intervals $[\widehat{L}_{\alpha/2}^{(\mathbb{E}\phi)}, +\infty)$ for $\mathbb{E}[\inf \phi(C(X))]$ and $(-\infty, \widehat{U}_{\alpha/2}^{(\mathbb{E}\phi)}]$ for $\mathbb{E}[\sup \phi(C(X))]$; i.e., $\widehat{L}_{\alpha/2}^{(\mathbb{E}\phi)}, \widehat{U}_{\alpha/2}^{(\mathbb{E}\phi)}$ such that

$$\mathbb{P}_{\widehat{L}_{\alpha/2}^{(\mathbb{E}\phi)}}\left[\widehat{L}_{\alpha/2}^{(\mathbb{E}\phi)} \leq \mathbb{E}[\inf \phi(C(X))]\right] \geq 1 - \frac{\alpha}{2}; \quad \mathbb{P}_{\widehat{U}_{\alpha/2}^{(\mathbb{E}\phi)}}\left[\mathbb{E}[\sup \phi(C(X))] \leq \widehat{U}_{\alpha/2}^{(\mathbb{E}\phi)}\right] \geq 1 - \frac{\alpha}{2}.$$

This can be readily done with off-the-shelf confidence intervals for the mean, such as CLT-based CIs, Hoeffding CIs and e-value-based methods (e.g. (Waudby-Smith & Ramdas, 2020)).

(iii) Produce the interval

$$\widehat{C}_{\alpha}^{(\mathbb{E}\phi)} := \left[\widehat{L}_{\alpha/2}^{(\mathbb{E}\phi)} - M \, \mathrm{Err}(C), \; \widehat{U}_{\alpha/2}^{(\mathbb{E}\phi)} + M \, \mathrm{Err}(C)\right]. \tag{1}$$

This is a simple procedure that benefits from good theoretical properties. In particular, the resulting interval is a valid $(1 - \alpha)$-confidence interval for $\mathbb{E}[\phi(Y)]$:

**Proposition 2.3.** *Under the conditions of Lemma 2.1, for any $\alpha \in (0, 1)$, let $\widehat{C}_{\alpha}^{(\mathbb{E}\phi)}$ be as in Equation 1. Then $\widehat{C}_{\alpha}^{(\mathbb{E}\phi)}$ is a valid $(1 - \alpha)$-confidence interval for $\mathbb{E}[\phi(Y)]$, i.e.,*

$$\mathbb{P}\left[\mathbb{E}[\phi(Y)] \in \widehat{C}_{\alpha}^{(\mathbb{E}\phi)}\right] \geq 1 - \alpha.$$

It is immediate to see that if the set predictor satisfies, e.g., privacy with regards to its calibration data, then so will the confidence interval $\widehat{C}^{(\mathbb{E}\phi)}$.[1] Similarly, if the conformal predictor is robust to outliers or strategic manipulations, so is the confidence interval.

We can also exactly quantify the size of $\widehat{C}_{\alpha}^{(\mathbb{E}\phi)}$ in terms of $M$, $\mathrm{Err}(C)$, the average predictive interval size and the tightness of the one-sided CIs for $\widehat{L}_{\alpha/2}^{(\mathbb{E}\phi)}$ and $\widehat{U}_{\alpha/2}^{(\mathbb{E}\phi)}$. Let $\mathrm{leb}$ be the Lebesgue measure and $\mathrm{hull}(A)$ the convex hull of $A$ (i.e., in $\mathbb{R}$ the smallest interval containing the set $A$). Then:

**Proposition 2.4.** *It holds that*

$$\mathrm{leb}\, \widehat{C}_{\alpha}^{(\mathbb{E}\phi)} = \mathbb{E}[\mathrm{leb}\, \mathrm{hull}(\phi(C(X)))] + 2M \, \mathrm{Err}(C)$$
$$+ (\mathbb{E}[\inf \phi(C(X))] - \widehat{L}_{\alpha/2}^{(\mathbb{E}\phi)}) + (\widehat{U}_{\alpha/2}^{(\mathbb{E}\phi)} - \mathbb{E}[\sup \phi(C(X))]).$$

From Proposition 2.4 it can be seen that our method works best with tight set predictors. As the set predictor approaches perfect accuracy – as is often the case in machine learning applications – the first two terms can be taken to approach zero. The last two terms, which concern the tightness of the one-sided confidence intervals $\widehat{L}_{\alpha/2}^{(\mathbb{E}\phi)}$ and $\widehat{U}_{\alpha/2}^{(\mathbb{E}\phi)}$, can be given an explicit form for specific methods for producing $\widehat{L}_{\alpha/2}^{(\mathbb{E}\phi)}$ and $\widehat{U}_{\alpha/2}^{(\mathbb{E}\phi)}$, but overall generally scale in order $O(n^{-1/2})$.

---

[1]If $(\epsilon, \delta)$-differential privacy is satisfied for the conformal calibration with relation to the calibration data, then our procedure amounts to post-processing atop the already-private set-predictor $C(\cdot)$, ands so our CI immediately satisfies $(\epsilon, \delta)$-differential privacy by the standard post-processing theorems of differential privacy.

## 2.2 Z-ESTIMATION AND M-ESTIMATION PROBLEMS

Going beyond means, we now consider the problem of Z-estimation, in which our estimand $\theta^\star \in \Theta$ (for some parameter space $\Theta$) is given as the solution to the estimating equation $\mathbb{E}_Y[\psi(Y; \theta^\star)] = 0$, for some function $\psi$. Z-estimation problems are common, with prominent examples being the inference of means (for $\psi(Y; \theta) = Y - \theta$), medians (for $\psi(Y; \theta) = \mathbb{1}[Y \leq \theta] - 0.5$), general quantiles (for the $q$-quantile, $\psi(Y; \theta) = \mathbb{1}[Y \leq \theta] - q$), regression coefficients (for $\psi((X, Y); \theta) = \theta X^2 - XY$) and more. Similar to how we have assumed bounded means in Section 2.1, we will assume here that $\psi(Y; \theta) \in [a_\theta, b_\theta]$ almost surely for each $\theta \in \Theta$, and let $M_\theta = b_\theta - a_\theta$. Again, for convenience, let $\psi(C(X); \theta) := \{\psi(y; \theta) : y \in C(X)\}$.

Consider the following procedure, which is close in spirit to the vanilla PPI procedure proposed by (Angelopoulos et al., 2023a): for each $\theta \in \Theta$, produce a lower one-sided $(1 - \alpha/2)$-confidence interval $[\widehat{L}^{(Z\psi)}_{\theta, \alpha/2}, +\infty)$ for $\mathbb{E}[\inf \psi(C(X); \theta)]$, and an upper one-sided $(1 - \alpha/2)$-confidence interval $(-\infty, \widehat{U}^{(Z\psi)}_{\theta, \alpha/2}]$ for $\mathbb{E}[\sup \psi(C(X); \theta)]$. Then, to estimate $\theta^\star$, produce the following set:

$$\widehat{C}^{(Z\psi)}_\alpha := \left\{ \theta \in \Theta : \widehat{L}^{(Z\psi)}_{\theta, \alpha/2} - M_\theta \operatorname{Err}(C) \leq 0 \leq \widehat{U}^{(Z\psi)}_{\theta, \alpha/2} + M_\theta \operatorname{Err}(C) \right\}. \tag{2}$$

By Lemma 2.1, it follows that this is a valid confidence interval for $\theta^\star$:

**Proposition 2.5.** *For any $\alpha \in (0, 1)$ let $\widehat{C}^{(Z\psi)}_\alpha$ be as in Equation 2. Then $\widehat{C}^{(Z\psi)}_\alpha$ is a valid $(1 - \alpha)$-confidence interval for $\theta^\star$, i.e.,*

$$\mathbb{P}\left[\theta^\star \in \widehat{C}^{(Z\psi)}_\alpha\right] \geq 1 - \alpha.$$

We can also bound the size of $\widehat{C}^{(Z\psi)}_\alpha$; however, due to its more implicit nature, this is more involved than the case of the inference of a mean in the previous section. Below we establish a result under the assumption that the one-sided confidence intervals are $K$-smooth in $\theta$ and that $\Theta$ is bounded.

**Proposition 2.6.** *Consider $\Theta \subset \mathbb{R}$ bounded by $B$ (i.e., for all $\theta, \theta' \in \Theta$, $\|\theta - \theta'\| \leq B$). Suppose that $\widehat{L}^{(Z\psi)}_{\theta, \alpha/2}$ and $\widehat{U}^{(Z\psi)}_{\theta, \alpha/2}$ are both $K$-smooth in $\theta$ (i.e., differentiable w.r.t. $\theta$, with $K$-Lipschitz derivative), $\widehat{L}^{(Z\psi)}_{\theta, \alpha/2} \leq \widehat{U}^{(Z\psi)}_{\theta, \alpha/2}$ and $M_\theta \leq M$ for all $\theta$ and that $\frac{\mathrm{d}}{\mathrm{d}\theta}\widehat{L}^{(Z\psi)}_{\theta^\star, \alpha/2}, \frac{\mathrm{d}}{\mathrm{d}\theta}\widehat{U}^{(Z\psi)}_{\theta^\star, \alpha/2} \neq 0$. Then*

$$\operatorname{leb} \widehat{C}^{(Z\psi)}_\alpha \leq \frac{1}{D_{\min}} \Bigg( \mathbb{E}[\operatorname{leb} \operatorname{hull}(\psi(C(X); \theta^\star))] + 2M \operatorname{Err}(C)$$

$$+ |\mathbb{E}[\inf \psi(C(X); \theta^\star)] - \widehat{L}^{(Z\psi)}_{\theta^\star, \alpha/2}| + |\widehat{U}^{(Z\psi)}_{\theta^\star, \alpha/2} - \mathbb{E}[\sup \psi(C(X); \theta^\star)]|$$

$$+ KB + \max\{a_{\theta^\star}, b_{\theta^\star}\} |1 - D_{\min}/D_{\max}| \Bigg),$$

*where $D_{\min} = \min\left\{ |\frac{\mathrm{d}}{\mathrm{d}\theta}\widehat{L}^{(Z\psi)}_{\theta^\star, \alpha/2}|, |\frac{\mathrm{d}}{\mathrm{d}\theta}\widehat{U}^{(Z\psi)}_{\theta^\star, \alpha/2}| \right\}$ and $D_{\max} = \max\left\{ |\frac{\mathrm{d}}{\mathrm{d}\theta}\widehat{L}^{(Z\psi)}_{\theta^\star, \alpha/2}|, |\frac{\mathrm{d}}{\mathrm{d}\theta}\widehat{U}^{(Z\psi)}_{\theta^\star, \alpha/2}| \right\}$.*

This means that the size of the resulting confidence interval is mainly governed by the average predictive interval size, $M$ and $\operatorname{Err}(C)$, and the tightness of the one-sided confidence intervals, as before, but now also takes into account how quickly $\psi$ passes through 0 at $\theta^\star$ (via the derivatives) and how "well-behaved" the one-sided confidence intervals are over $\Theta$.

In the case of inference of a mean, where $\psi(Y; \theta) = \phi(Y) - \theta$ with sufficiently regular methods for obtaining the one-sided confidence intervals (e.g. CLT-based CIs or Hoeffding bounds), the derivatives will equal one everywhere (i.e., $D_{\min} = D_{\max} = 1$) and the one-sided confidence intervals will be 0-smooth (i.e., $K = 0$), and we recover Proposition 2.4 except for the modulus in the terms concerning the tightness of the one-sided CIs.

A similar procedure is also applicable to M-estimation problems, in which we want to infer $\theta^\star = \arg\min_{\theta \in \Theta} \mathbb{E}_Y[\ell(Y; \theta)]$ with $\ell$ (sub)differentiable in $\theta$. Much like Z-estimation, M-estimation problems are broadly applicable, including not only means, quantiles and regression coefficients but also more involved estimands such as robust statistics, maximum likelihood estimates with nonlinear models and more. For boundedness, we make the assumption that $\ell'(Y; \theta) \subset [a_\theta, b_\theta]$ almost surely for all $\theta \in \Theta$, and let $M_\theta = b_\theta - a_\theta$ for convenience.

Since the loss is differentiable, the minimum $\theta^\star$ occurs in a point where $\mathbb{E}[\frac{\mathrm{d}}{\mathrm{d}\theta}\ell(Y;\theta^\star)] = 0$ (if $\ell$ is furthermore convex in $\theta$, then the two are equivalent). This thus reduces the M-estimation problem to a Z-estimation one, which we can solve: for each $\theta \in \Theta$, produce lower and upper $(1 - \alpha/2)$-confidence intervals $[\widehat{L}^{(\mathrm{M}\ell)}_{\theta,\alpha/2}, +\infty)$ and $(-\infty, \widehat{U}^{(\mathrm{M}\ell)}_{\theta,\alpha/2}]$ for $\mathbb{E}[\inf \frac{\mathrm{d}}{\mathrm{d}\theta}\ell(C(X);\theta^\star)]$ and $\mathbb{E}[\sup \frac{\mathrm{d}}{\mathrm{d}\theta}\ell(C(X);\theta^\star)]$, respectively, and produce the set

$$\widehat{C}^{(\mathrm{M}\ell)}_\alpha := \left\{ \theta \in \Theta : \widehat{L}^{(\mathrm{M}\ell)}_{\theta,\alpha/2} - M_\theta \operatorname{Err}(C) \leq 0 \leq \widehat{U}^{(\mathrm{M}\ell)}_{\theta,\alpha/2} + M_\theta \operatorname{Err}(C) \right\}. \tag{3}$$

This is a valid $(1 - \alpha)$ confidence interval for $\theta^\star$:

**Proposition 2.7.** *For any $\alpha \in (0, 1)$, let $\widehat{C}^{(\mathrm{M}\ell)}_\alpha$ be as in Equation 3. Then $\widehat{C}^{(\mathrm{M}\ell)}_\alpha$ is a valid confidence interval for $\theta^\star$, i.e.,*

$$\mathbb{P}\left[ \theta^\star \in \widehat{C}^{(\mathrm{M}\ell)}_\alpha \right] \geq 1 - \alpha.$$

We can also similarly bound the size of the resulting confidence interval, which now looks at the steepness of the one-sided intervals for the derivatives, i.e., the curvature of $\ell$ around $\theta^\star$; see Theorem A.7 in the appendix.

### 2.3 INFERENCE WITH E-VALUES

Following the work of (Csillag et al., 2025), we now extend our set of inference tasks to those powered by e-values, a modern and enticing alternative to p-values (Ramdas et al., 2022; Ramdas & Wang, 2024). An e-value for a null hypothesis $H_0$ is an nonnegative real random variable $E$ such that if $H_0$ holds then $\mathbb{E}[E] \leq 1$ (and ideally $\mathbb{E}[E] \gg 1$ otherwise). By Markov's inequality, it is unlikely that the e-value achieves a high value under the null ($\mathbb{P}[E > a] \leq \mathbb{E}[E]/a \leq 1/a$), and so a high e-value provides evidence against the null. Furthermore, e-values satisfy many desirable properties missed by p-values while being highly versatile; we refer the interested reader to (Ramdas et al., 2022) and (Ramdas & Wang, 2024) for an introduction.

Consider the problem of testing a null hypothesis $H_0$. Let $E_n$ be an e-value with a test supermartingale structure, which can be written in the form $E_n := \prod_{i=1}^n e_i(Y_i)$ for a predictable sequence $(e_i)_{i=1}^\infty$ of 'components' of the e-value; i.e. each $e_i$ can be arbitrarily dependent on the samples before time $i$ (but nothing else). Analogous to the previous sections, we will also require a boundedness condition, in that for all $i$, $e_i(Y) \in [a_i, b_i]$ almost surely for some predictable sequences $(a_i)_{i=1}^\infty$ and $(b_i)_{i=1}^\infty$, and with $a_i > 0$ for all $i$. These boundedness conditions can be enforced by simple rescaling and clipping, albeit at a slight loss of power.

With a possibly-moving predictable sequence of conformal predictors $(C_i)_{i=1}^\infty$ in hand, the conformal prediction-powered e-value can be constructed as follows:

$$E^{\mathrm{ppi}-(C)}_n := \prod_{i=1}^n \operatorname{rescale}_{\eta_i}\left( \inf e_i(C_i(X_i)) - (b_i - a_i)\operatorname{Err}(C_i) \right), \tag{4}$$

where $(\eta_i)_{i=1}^\infty$ is a predictable sequence with $0 \leq \eta_i \leq (1 - a_i - (b_i - a_i)\operatorname{Err}(C_i))^{-1}$ for all $i = 1, 2, \ldots$, $\operatorname{rescale}_\eta(e) = 1 + \eta(e - 1)$ and $e_i(C_i(X_i)) = \{e_i(y) : y \in C_i(X_i)\}$ for convenience. The sequence $(\eta_i)$ is analogous to the bets usually present in e-values from the testing by betting literature, cf. (Shafer, 2021; Waudby-Smith & Ramdas, 2020; Ramdas et al., 2022); it ensures that the e-values remain nonnegative as well as allowing for gains in power, e.g. when the $\eta$s are chosen to approximately maximize the e-value's growth rate. It follows that $E^{\mathrm{ppi}-(C)}_n$ is a valid e-value, inheriting the test supermartingale structure of $E_n$.

**Proposition 2.8.** *Let $E^{\mathrm{ppi}-(C)}_n$ be as in Equation 4. Then $(E^{\mathrm{ppi}-(C)}_1, E^{\mathrm{ppi}-(C)}_2, \ldots)$ is a test supermartingale, and $E^{\mathrm{ppi}-(C)}_\tau$ is an e-value for any stopping time $\tau$.*

We can also analyze the power of our e-values. The natural way of measuring the power of an e-value is by the means of its expected growth rate (Kelly, 1956). For conformal prediction-powered e-values, it will be close to that of the original e-value as long as the conformal predictive sets are sufficiently small and with a low Err.

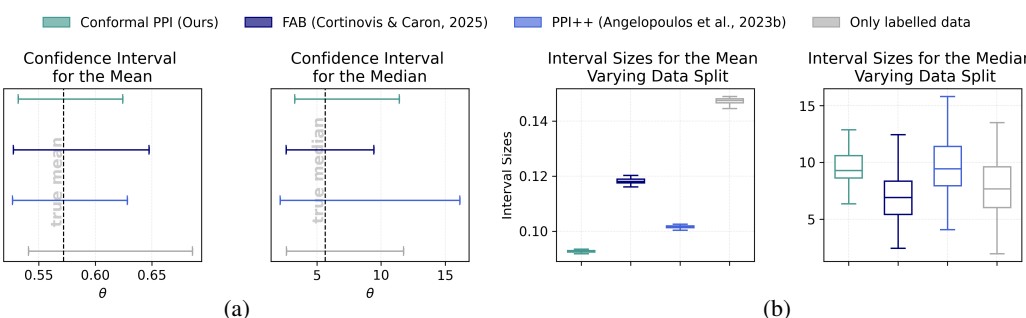

Figure 1: **Our method is comparable to existing prediction-powered procedures.** We conduct experiments on two datasets where previous prediction-powered methods are applicable: one on the prevalence of phishing attacks (a mean), and another on characterizing gene expression levels (a median). In (a) we see one realization of our CIs along with baselines, while in (b) we analyze the distribution of the interval sizes over varying data splits. In both cases our procedure outperforms only using labelled data, while edging over prior methods for the mean estimation task.

**Proposition 2.9.** *If $e_i(\cdot) \in [a_i, b_i]$ for every $i$, then there exists some constant $r > 0$ independent of $n$ for which*

$$\mathbb{E}\left[\frac{1}{n}\log E_n^{\mathrm{ppi}-(C)}\right] \geq \mathbb{E}\left[\frac{1}{n}\log E_n\right] - \frac{1}{n}\sum_{i=1}^{n}\mathbb{E}[\mathrm{leb}\,\mathrm{hull}(\log e_i(C_i(X_i)))]$$

$$- \frac{r}{n}\sum_{i=1}^{n}\mathbb{E}\left[h_i(\eta_i)\mathrm{Err}(C_i)\right] - \frac{r}{n}\sum_{i=1}^{n}\mathbb{E}\big[|1 - \eta_i|\,|\inf e_i(C_i(X_i)) - 1|\big],$$

*where $h_i(\eta_i) = \log\frac{b_i}{a_i} + \eta_i(b_i - a_i)$, which is increasing in $\eta_i$.*

Proposition 2.9 makes apparent a trade-off in the choice of the $(\eta_i)_{i=1}^{\infty}$: by choosing a lower $\eta_i$ we reduce the effect of the $(b_i - a_i)\mathrm{Err}(C_i)$ penalty on the e-values, but incur a slight loss in power due to the rescaling. An optimal balance can be struck by choosing log-optimal $(\eta_i)_{i=1}^{\infty}$, as is usual in the testing by betting literature.

These e-values can also be directly used for confidence intervals/sequences and general e-value-based procedures; see Appendix B.1.

## 3  EXPERIMENTS AND CASE STUDIES

To empirically assess our method, we devise a series of experiments on real-world datasets. We first consider the estimation of means and quantiles, in which we can compare our approach to previous methods for prediction-powered inference (Section 3.1). We then turn to more elaborate scenarios, which our procedure naturally solves but were out of reach for previous methods: first for prediction-powered inference with private labelled data (Section 3.2) and then for prediction-powered anytime-valid hypothesis testing on time series sans active data collection (Section 3.3). Experiment details can be found in Appendix C.

Code for all experiments can be found on `[redacted URL]` (present in the supplementary material). All experiments were run on an AMD Ryzen 9 5950X CPU, with 64GB of RAM.

### 3.1  COMPARISON WITH PREVIOUS PREDICTION-POWERED INFERENCE METHODS

We consider two inferential tasks: estimating the prevalence of phishing websites (which is a probability, and thus a mean), and the inference of gene expression levels, as measured by their quantiles (in particular, a median). Phishing is one of the most common types of cybercrime, and quantifying the prevalence of phishing domains allows cybersecurity firms and ISPs to gauge the scale of the problem and allocate resources to prevent these attacks. As for gene expression levels, these can

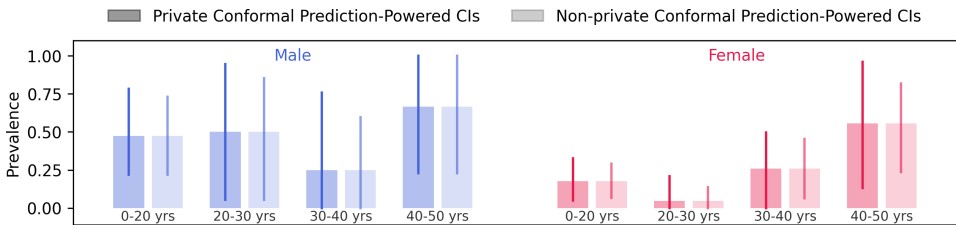

Figure 2: **Conformal prediction-powered inference with differential privacy.** We apply our method to analyze the recurrence of thyroid cancer atop private patient data. With a single inference-agnostic and differentially-private calibration, we are able to do prediction-powered inference for the probabilities of recurrence for various strata, with minimal increase in interval size compared to a non-private calibration.

be used to better understand cis-regulation in humans, which is important to the study of complex diseases. As is usual in the prediction-powered inference literature, we evaluate our procedures on large labelled datasets, namely those of (Mohammad & McCluskey, 2012) and (Vaishnav et al., 2022) for the phishing and gene expression level tasks, respectively.

For the phishing dataset, we allocate most of the data for training a predictive model; for the gene expression dataset, we use the predictions from the readily available model of (Vaishnav et al., 2022). We then split the remaining data between a large test set (where we discard the labels $Y$) and a smaller calibration set (for which we will use both $X$ and $Y$). On this calibration set, we perform split conformal prediction to obtain a calibrated set-predictor, using the conformity score $(x, y) \mapsto -\widehat{p}(y \mid x)$ for the phishing dataset and $(x, y) \mapsto |\widehat{\mu}(x) - y|$ for the gene expression dataset,[2] where $\widehat{p}$ and $\widehat{\mu}$ denote the respective predictive models.

We compare four methods. **Conformal PPI (Ours):** we use the conformal predictors fit on the calibration set, and compute our conformal prediction-powered CIs on the test set as outlined in Sections 2.1 and 2.2. **PPI++ (Angelopoulos et al., 2023b):** the calibration set is used in conjunction with the test set to form an unbiased estimate of the loss of an M-estimator, with a data-dependent 'power tuning' parameter $\lambda$. Asymptotic analysis then allows for the construction of valid CIs. **FAB (Cortinovis & Caron, 2025):** FAB extends PPI/PPI++ by introducing a prior over the quality of the predictive model. It provides tighter CIs when the observed prediction quality is likely under the prior, while ensuring graceful degradation otherwise (for well-chosen priors, e.g., horseshoe prior). **Only labelled samples:** we compute a classical CI using the calibration data, ignoring the test set.

Figure 1 shows these procedures in action. In particular we showcase instances of our confidence intervals for the mean and median of the labels of our datasets, along with the distribution of their interval sizes over varying data split seeds. Our approach is competitive with previous methods, beating the intervals that use only the labelled samples. In the case of the mean, our method in fact provides the tightest confidence intervals. For the median ours is not as tight as FAB (Cortinovis & Caron, 2025), but surpasses PPI++ (Angelopoulos et al., 2023b). We also note that our method achieves the smallest variance.

### 3.2 PREDICTION-POWERED INFERENCE WITH PRIVATE LABELLED DATA

In this section we illustrate the use of our method for analyzing the recurrence of thyroid cancer. As with many medical applications, access to medical records is required. Due to their sensitive nature, all labelled data must be treated in a differentially private manner; this is beyond the scope of previous prediction-powered procedures, which do not satisfy differential privacy and thus may leak information.

We use the dataset of (Borzooei & Tarokhian, 2023), which contains readily accessible clinical data (e.g. from surveys), along with an indicator of whether the patient's cancer recurred. We split this dataset into training, calibration and test sets. In the training set, we fit a model to predict the recurrence of thyroid cancer. The calibration set is then used for the differentially private conformal prediction method of (Angelopoulos et al., 2021), using the conformity score $(x, y) \mapsto \widehat{p}(y \mid x)$.

---

[2]The pretrained model only predicts $\widehat{\mu}(x)$, so we cannot use a more adaptive score (such as conformalized quantile regression).

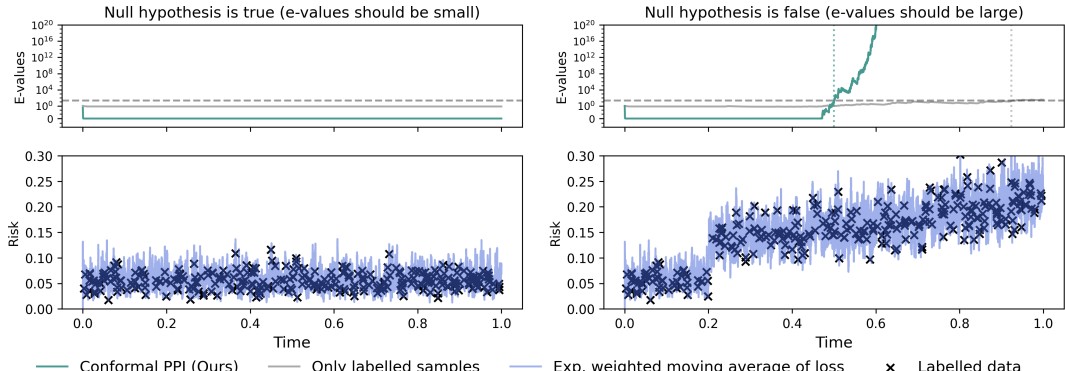

Figure 3: **Conformal prediction-powered continuous risk monitoring.** Our method can also be naturally applied for continuous risk monitoring by simply using online conformal prediction methods for the calibration. It satisfies strong anytime-valid guarantees in a dynamic setting without requiring active data collection. The resulting procedure rejects nulls much more quickly than simply using labelled samples, attaining high statistical power.

Finally, we use this conformal predictor to perform several inferences on the test set, estimating the probabilities of recurrence for different strata of the population. The results can be seen in Figure 2; we find that the differentially private calibration yields only minor increases of the interval sizes while vastly increasing safety.

Also worth highlighting is that our procedure allows us to use a single private calibration for multiple inferences, which can even be defined post-hoc. This is in contrast to previous prediction-powered methods, which require access to the calibration data for every inference, potentially compromising privacy.

### 3.3 PREDICTION-POWERED RISK MONITORING VIA ONLINE CONFORMAL PREDICTION

Consider the task of tracking the risk of a deployed model on-line, so that we ensure it never goes past some determined safety level. In this setting, continuously receive inputs for our predictive model, but only occasionally receive labels that would allow us to assess the correctness of our predictions. This is a problem with significant temporal structure, putting it out-of-reach of most prediction-powered methods (which can only handle static i.i.d. settings). As far as we are aware the only applicable method is that of (Csillag et al., 2025), but it requires an active data collection regime; ours is trivially applicable to an observational regime.

The task can be framed as an anytime-valid test for the null hypothesis that the risk is within the safety level at all times; such a hypothesis test can then be done using, for example, the e-value framework of (Podkopaev & Ramdas, 2021).

For our experiment, we use the dataset of (Blackard, 1998) for forest cover type prediction. We create two versions of the dataset: the original one, in which the null hypothesis holds (i.e., no distribution shift), and another one in which we increasingly poison the data by selecting harder samples with increasing probability past a change-point, rendering the null hypothesis false.

Each version of the dataset is partitioned into training, validation, and test splits. A predictive model is fit on the training data, whose loss we then estimate on the validation set. Our desired safety level is then taken to be this validation loss plus a small tolerance threshold. Still on the validation set, we train an auxiliary model to infer the predictive model's residuals. Finally, on the test set we monitor the on-line risk: we use our occasional labelled samples for the online conformal prediction method of (Angelopoulos et al., 2024) atop the auxiliary model, and use the resulting set predictor for our conformal prediction-powered e-values. The $(\eta_i)_{i=1}^{\infty}$ are chosen to approximately maximize the growth rate (cf. Appendix C.4).

Figure 3 shows the results of our experiment, comparing it to only using the occasional labelled samples. When the null is false, our prediction-powered e-values reject it much more quickly and confidently than only using the labelled data, while guaranteeing a low false positive rate.

## 4 CONCLUSION

In this paper, we established a general connection between prediction-powered inference and conformal prediction, enabling prediction-powered methods with additional guarantees like privacy and robustness. Our framework leverages calibrated conformal set-predictors to inherit rich properties from the conformal literature, overcoming the case-specific limitations of previous work and opening new practical applications previously out of reach. Beyond being readily applicable to diverse practical settings, we believe our framework establishes an important baseline for future research on prediction-powered inference with additional guarantees.

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

## A  THEOREMS AND PROOFS

**Lemma A.1** (Lemma 2.1 in the main text). *Let $C : \mathcal{X} \to 2^{\mathcal{Y}}$ be a set predictor and suppose that $\phi(Y) \in [a, b]$ almost surely. Then*

$$\mathbb{E}[\inf \phi(C(X))] - M \operatorname{Err}(C) \leq \mathbb{E}[\phi(Y)] \leq \mathbb{E}[\sup \phi(C(X))] + M \operatorname{Err}(C).$$

*Proof.* We will show this in two parts:

(i) $\mathbb{E}[\inf \phi(C(X))] - M \operatorname{Err}(C) \leq \mathbb{E}[\phi(Y)]$, by showing that $\mathbb{E}[\inf \phi(C(X)) - \phi(Y)] \leq M \operatorname{Err}(C)$;

(ii) $\mathbb{E}[\phi(Y)] \leq \mathbb{E}[\sup \phi(C(X))] + M \operatorname{Err}(C)$, by showing that $\mathbb{E}[\phi(Y) - \sup \phi(C(X))] \leq M \operatorname{Err}(C)$.

For (i), by the law of total expectation:

$$\mathbb{E}[\inf \phi(C(X)) - \phi(Y)] = \mathbb{E}[\inf \phi(C(X)) - \phi(Y)|Y \in C(X)] \, \mathbb{P}[Y \in C(X)]$$
$$+ \mathbb{E}[\inf \phi(C(X)) - \phi(Y)|Y \notin C(X)] \, \mathbb{P}[Y \notin C(X)];$$

Now, given that $Y \in C(X)$, it must hold that $\phi(Y) \in \phi(C(X))$, and so $\inf \phi(C(X)) \leq \phi(Y)$; thus $\mathbb{E}[\inf \phi(C(X)) - \phi(Y)|Y \in C(X)] \leq 0$. Additionally, note that because both $\phi(C(X))$ and $\phi(Y)$ are bounded in $[a, b]$, it holds that $\inf \phi(C(X)) - \phi(Y) \leq b - a = M$ almost surely, and so $\mathbb{E}[\inf \phi(C(X)) - \phi(Y)|Y \notin C(X)] \leq M$. Thus

$$\mathbb{E}[\inf \phi(C(X)) - \phi(Y)] \leq 0 + M \operatorname{Err}(C) = M \operatorname{Err}(C).$$

The upper bound (ii) follows analogously: by the law of total expectation,

$$\mathbb{E}[\phi(Y) - \sup \phi(C(X))] = \mathbb{E}[\phi(Y) - \sup \phi(C(X))|Y \in C(X)] \, \mathbb{P}[Y \in C(X)]$$
$$+ \mathbb{E}[\phi(Y) - \sup \phi(C(X))|Y \notin C(X)] \, \mathbb{P}[Y \notin C(X)];$$

Now, given that $Y \in C(X)$, it must hold that $\phi(Y) \in \phi(C(X))$, and so $\phi(Y) \leq \sup \phi(C(X))$; thus $\mathbb{E}[\phi(Y) - \sup \phi(C(X))|Y \in C(X)] \leq 0$. Additionally, note that because both $\phi(C(X))$ and $\phi(Y)$ are bounded in $[a, b]$, it holds that $\phi(Y) - \sup \phi(C(X)) \leq b - a = M$ almost surely, and so $\mathbb{E}[\phi(Y) - \sup \phi(C(X))|Y \notin C(X)] \leq M$. Thus

$$\mathbb{E}[\phi(Y) - \sup \phi(C(X))] \leq 0 + M \operatorname{Err}(C) = M \operatorname{Err}(C),$$

and we conclude. $\qquad \square$

**Proposition A.2** (Proposition 2.3 in the main text). *Under the conditions of Lemma 2.1, for any $\alpha \in (0, 1)$, let $\widehat{C}_\alpha^{(\mathbb{E}\phi)}$ be as in Equation 3 from the main text. Then $\widehat{C}_\alpha^{(\mathbb{E}\phi)}$ is a valid $(1 - \alpha)$-confidence interval for $\mathbb{E}[\phi(Y)]$, i.e.,*

$$\mathbb{P}\left[\mathbb{E}[\phi(Y)] \in \widehat{C}_\alpha^{(\mathbb{E}\phi)}\right] \geq 1 - \alpha.$$

*Proof.*

$$\mathbb{P}\left[\mathbb{E}[\phi(Y)] \notin \widehat{C}_\alpha^{(\mathbb{E}\phi)}\right] = \mathbb{P}\left[\widehat{L}_{\alpha/2}^{(\mathbb{E}\phi)} - M \operatorname{Err}(C) \not\leq \mathbb{E}[\phi(Y)] \text{ or } \mathbb{E}[\phi(Y)] \not\leq \widehat{U}_{\alpha/2}^{(\mathbb{E}\phi)} + M \operatorname{Err}(C)\right]$$

$$\leq \mathbb{P}\left[\widehat{L}_{\alpha/2}^{(\mathbb{E}\phi)} - M \operatorname{Err}(C) \not\leq \mathbb{E}[\phi(Y)]\right] + \mathbb{P}\left[\mathbb{E}[\phi(Y)] \not\leq \widehat{U}_{\alpha/2}^{(\mathbb{E}\phi)} + M \operatorname{Err}(C)\right]$$

$$= \mathbb{P}\left[\widehat{L}_{\alpha/2}^{(\mathbb{E}\phi)} \not\leq \mathbb{E}[\phi(Y)] + M \operatorname{Err}(C)\right] + \mathbb{P}\left[\mathbb{E}[\phi(Y)] - M \operatorname{Err}(C) \not\leq \widehat{U}_{\alpha/2}^{(\mathbb{E}\phi)}\right]$$

$$\leq \mathbb{P}\left[\widehat{L}_{\alpha/2}^{(\mathbb{E}\phi)} \not\leq \mathbb{E}[\inf \phi(C(X))]\right] + \mathbb{P}\left[\mathbb{E}[\sup \phi(C(X))] \not\leq \widehat{U}_{\alpha/2}^{(\mathbb{E}\phi)}\right],$$

and, since $\widehat{L}_{\alpha/2}^{(\mathbb{E}\phi)}$ and $\widehat{U}_{\alpha/2}^{(\mathbb{E}\phi)}$ are one-sided confidence intervals, it follows that

$$\mathbb{P}\left[\widehat{L}_{\alpha/2}^{(\mathbb{E}\phi)} \not\leq \mathbb{E}[\inf \phi(C(X))]\right] + \mathbb{P}\left[\mathbb{E}[\sup \phi(C(X))] \not\leq \widehat{U}_{\alpha/2}^{(\mathbb{E}\phi)}\right] \leq \frac{\alpha}{2} + \frac{\alpha}{2} = \alpha. \qquad \square$$

**Proposition A.3** (Proposition 2.4 in the main text). *It holds that*

$$\operatorname{leb} \widehat{C}_\alpha^{(\mathbb{E}\phi)} = \mathbb{E}[\operatorname{leb} \operatorname{hull}(\phi(C(X)))] + 2M \operatorname{Err}(C)$$
$$+ (\mathbb{E}[\inf \phi(C(X))] - \widehat{L}_{\alpha/2}^{(\mathbb{E}\phi)}) + (\widehat{U}_{\alpha/2}^{(\mathbb{E}\phi)} - \mathbb{E}[\sup \phi(C(X))]).$$

*Proof.*

$$\text{leb}\,\widehat{C}_{\alpha}^{(\mathbb{E}\phi)} = \left(\widehat{U}_{\alpha/2}^{(\mathbb{E}\phi)} + M\,\text{Err}(C)\right) - \left(\widehat{L}_{\alpha/2}^{(\mathbb{E}\phi)} - M\,\text{Err}(C)\right)$$

$$= \widehat{U}_{\alpha/2}^{(\mathbb{E}\phi)} - \widehat{L}_{\alpha/2}^{(\mathbb{E}\phi)} + 2M\,\text{Err}(C)$$

$$= \left(\mathbb{E}[\sup\phi(C(X))] + \widehat{U}_{\alpha/2}^{(\mathbb{E}\phi)} - \mathbb{E}[\sup\phi(C(X))]\right)$$

$$\quad - \left(\mathbb{E}[\inf\phi(C(X))] + \widehat{L}_{\alpha/2}^{(\mathbb{E}\phi)} - \mathbb{E}[\inf\phi(C(X))]\right) + 2M\,\text{Err}(C)$$

$$= (\mathbb{E}[\sup\phi(C(X))] - \mathbb{E}[\inf\phi(C(X))]) + 2M\,\text{Err}(C)$$

$$\quad + \left(\widehat{U}_{\alpha/2}^{(\mathbb{E}\phi)} - \mathbb{E}[\sup\phi(C(X))]\right) - \left(\widehat{L}_{\alpha/2}^{(\mathbb{E}\phi)} - \mathbb{E}[\inf\phi(C(X))]\right)$$

$$= \mathbb{E}[\text{leb}\,\text{hull}(\phi(C(X)))] + 2M\,\text{Err}(C)$$

$$\quad + \left(\widehat{U}_{\alpha/2}^{(\mathbb{E}\phi)} - \mathbb{E}[\sup\phi(C(X))]\right) + \left(\mathbb{E}[\inf\phi(C(X))] - \widehat{L}_{\alpha/2}^{(\mathbb{E}\phi)}\right). \qquad \square$$

**Proposition A.4** (Proposition 2.5 in the main text)**.** *For any $\alpha \in (0,1)$ let $\widehat{C}_{\alpha}^{(\mathbb{Z}\psi)}$ be as in Equation 4 from the main text. Then $\widehat{C}_{\alpha}^{(\mathbb{Z}\psi)}$ is a valid $(1-\alpha)$-confidence interval for $\theta^{\star}$, i.e.,*

$$\mathbb{P}\left[\theta^{\star} \in \widehat{C}_{\alpha}^{(\mathbb{Z}\psi)}\right] \geq 1 - \alpha.$$

*Proof.*

$$\mathbb{P}\left[\theta^{\star} \notin \widehat{C}_{\alpha}^{(\mathbb{Z}\psi)}\right] = \mathbb{P}\left[\widehat{L}_{\theta^{\star},\alpha/2}^{(\mathbb{Z}\psi)} - M_{\theta^{\star}}\,\text{Err}(C) \not\leq 0 \text{ or } 0 \not\leq \widehat{U}_{\theta^{\star},\alpha/2}^{(\mathbb{Z}\psi)} + M_{\theta^{\star}}\,\text{Err}(C)\right]$$

$$\leq \mathbb{P}\left[\widehat{L}_{\theta^{\star},\alpha/2}^{(\mathbb{Z}\psi)} - M_{\theta^{\star}}\,\text{Err}(C) \not\leq 0\right] + \mathbb{P}\left[0 \not\leq \widehat{U}_{\theta^{\star},\alpha/2}^{(\mathbb{Z}\psi)} + M_{\theta^{\star}}\,\text{Err}(C)\right];$$

Now, by definition $\mathbb{E}[\psi(Y;\theta^{\star})] = 0$, and so the above is equivalent to

$$\mathbb{P}\left[\widehat{L}_{\theta^{\star},\alpha/2}^{(\mathbb{Z}\psi)} - M_{\theta^{\star}}\,\text{Err}(C) \not\leq \mathbb{E}[\psi(Y;\theta^{\star})]\right] + \mathbb{P}\left[\mathbb{E}[\psi(Y;\theta^{\star})] \not\leq \widehat{U}_{\theta^{\star},\alpha/2}^{(\mathbb{Z}\psi)} + M_{\theta^{\star}}\,\text{Err}(C)\right]$$

$$= \mathbb{P}\left[\widehat{L}_{\theta^{\star},\alpha/2}^{(\mathbb{Z}\psi)} \not\leq \mathbb{E}[\psi(Y;\theta^{\star})] + M_{\theta^{\star}}\,\text{Err}(C)\right] + \mathbb{P}\left[\mathbb{E}[\psi(Y;\theta^{\star})] - M_{\theta^{\star}}\,\text{Err}(C) \not\leq \widehat{U}_{\theta^{\star},\alpha/2}^{(\mathbb{Z}\psi)}\right]$$

$$\leq \mathbb{P}\left[\widehat{L}_{\theta^{\star},\alpha/2}^{(\mathbb{Z}\psi)} \not\leq \mathbb{E}[\inf\psi(C(X);\theta^{\star})]\right] + \mathbb{P}\left[\mathbb{E}[\sup\psi(C(X);\theta^{\star})] \not\leq \widehat{U}_{\theta^{\star},\alpha/2}^{(\mathbb{Z}\psi)}\right],$$

and since the $\widehat{L}_{\theta^{\star},\alpha/2}^{(\mathbb{Z}\psi)}$ and $\widehat{U}_{\theta^{\star},\alpha/2}^{(\mathbb{Z}\psi)}$ are one-sided confidence intervals, it holds that

$$\mathbb{P}\left[\widehat{L}_{\theta^{\star},\alpha/2}^{(\mathbb{Z}\psi)} \not\leq \mathbb{E}[\inf\psi(C(X);\theta^{\star})]\right] + \mathbb{P}\left[\mathbb{E}[\sup\psi(C(X);\theta^{\star})] \not\leq \widehat{U}_{\theta^{\star},\alpha/2}^{(\mathbb{Z}\psi)}\right] \leq \frac{\alpha}{2} + \frac{\alpha}{2} = \alpha.$$

$$\square$$

**Proposition A.5** (Proposition 2.6 in the main text)**.** *Consider $\Theta \subset \mathbb{R}$ bounded by $B$ (i.e., for all $\theta, \theta' \in \Theta$, $\|\theta - \theta'\| \leq B$). Suppose that $\widehat{L}_{\theta,\alpha/2}^{(\mathbb{Z}\psi)}$ and $\widehat{U}_{\theta,\alpha/2}^{(\mathbb{Z}\psi)}$ are both $K$-smooth in $\theta$ (i.e., differentiable w.r.t. $\theta$, with $K$-Lipschitz derivative), $\widehat{L}_{\theta,\alpha/2}^{(\mathbb{Z}\psi)} \leq \widehat{U}_{\theta,\alpha/2}^{(\mathbb{Z}\psi)}$ and $M_{\theta} \leq M$ for all $\theta$ and that $\frac{\mathrm{d}}{\mathrm{d}\theta}\widehat{L}_{\theta^{\star},\alpha/2}^{(\mathbb{Z}\psi)}, \frac{\mathrm{d}}{\mathrm{d}\theta}\widehat{U}_{\theta^{\star},\alpha/2}^{(\mathbb{Z}\psi)} \neq 0$. Then*

$$\text{leb}\,\widehat{C}_{\alpha}^{(\mathbb{Z}\psi)} \leq \frac{1}{D_{\min}}\Bigg(\mathbb{E}[\text{leb}\,\text{hull}(\psi(C(X);\theta^{\star}))] + 2M\,\text{Err}(C)$$

$$\qquad + |\mathbb{E}[\inf\psi(C(X);\theta^{\star})] - \widehat{L}_{\theta^{\star},\alpha/2}^{(\mathbb{Z}\psi)}| + |\widehat{U}_{\theta^{\star},\alpha/2}^{(\mathbb{Z}\psi)} - \mathbb{E}[\sup\psi(C(X);\theta^{\star})]|$$

$$\qquad + KB + \max\{a_{\theta^{\star}}, b_{\theta^{\star}}\}\,|1 - D_{\min}/D_{\max}|\Bigg),$$

*where $D_{\min} = \min\left\{|\frac{\mathrm{d}}{\mathrm{d}\theta}\widehat{L}_{\theta^{\star},\alpha/2}^{(\mathbb{Z}\psi)}|, |\frac{\mathrm{d}}{\mathrm{d}\theta}\widehat{U}_{\theta^{\star},\alpha/2}^{(\mathbb{Z}\psi)}|\right\}$ and $D_{\max} = \max\left\{|\frac{\mathrm{d}}{\mathrm{d}\theta}\widehat{L}_{\theta^{\star},\alpha/2}^{(\mathbb{Z}\psi)}|, |\frac{\mathrm{d}}{\mathrm{d}\theta}\widehat{U}_{\theta^{\star},\alpha/2}^{(\mathbb{Z}\psi)}|\right\}$.*

*Proof.* For convenience, let $u(\theta) = \widehat{U}_{\theta,\alpha/2}^{(Z\psi)}$ and $\ell(\theta) = \widehat{L}_{\theta,\alpha/2}^{(Z\psi)}$.

We will do a first-order expansion around $\theta^\star$. Thanks to the $K$-smoothness assumption, it holds that, for all $\theta \in \Theta$,

$$u(\theta) + M_\theta \mathrm{Err}(C) \leq u(\theta) + M\mathrm{Err}(C) \leq u(\theta^\star) + M\mathrm{Err}(C) + u'(\theta^\star)(\theta - \theta^\star) + \frac{K}{2}\|\theta - \theta^\star\|^2$$

$$\leq u(\theta^\star) + M\mathrm{Err}(C) + u'(\theta^\star)(\theta - \theta^\star) + \frac{KB}{2};$$

(5)

$$\ell(\theta) - M_\theta \mathrm{Err}(C) \geq \ell(\theta) - M\mathrm{Err}(C) \geq \ell(\theta^\star) - M\mathrm{Err}(C) + \ell'(\theta^\star)(\theta - \theta^\star) - \frac{K}{2}\|\theta - \theta^\star\|^2$$

$$\geq \ell(\theta^\star) - M\mathrm{Err}(C) + \ell'(\theta^\star)(\theta - \theta^\star) - \frac{KB}{2}.$$

(6)

Consider then the set

$$S := \left\{ \theta \in \mathbb{R} : \ell(\theta^\star) - M\mathrm{Err}(C) + \ell'(\theta^\star)(\theta - \theta^\star) - \frac{KB}{2} \right.$$

$$\left. \leq 0 \leq u(\theta^\star) + M\mathrm{Err}(C) + u'(\theta^\star)(\theta - \theta^\star) + \frac{KB}{2} \right\}.$$

By Equations 6 and 5, it must hold that $\widehat{C}_\alpha^{(Z\psi)} \subset S$, and thus $\mathrm{leb}\,\widehat{C}_\alpha^{(Z\psi)} \leq \mathrm{leb}\,S$.

$S$ has a much more amenable form thanks to the first-order expansion, which allows us to quantify its measure precisely. First, note that $S$ is a convex subset of $\mathbb{R}$, and thus an interval. So all that we need to do is to find its endpoints, which can be done by solving its constraints for their zeros (which, since the derivatives at $\theta^\star$ are not nil, must be unique).

$$\ell(\theta^\star) - M\mathrm{Err}(C) + \ell'(\theta^\star)(\theta - \theta^\star) - \frac{KB}{2} = 0$$

$$\iff \ell'(\theta^\star)(\theta - \theta^\star) = \frac{KB}{2} + M\mathrm{Err}(C) - \ell(\theta^\star)$$

$$\iff \theta - \theta^\star = \frac{KB/2 + M\mathrm{Err}(C) - \ell(\theta^\star)}{\ell'(\theta^\star)}$$

$$\iff \theta = \theta^\star + \frac{KB/2 + M\mathrm{Err}(C) - \ell(\theta^\star)}{\ell'(\theta^\star)};$$

and

$$u(\theta^\star) + M\mathrm{Err}(C) + u'(\theta^\star)(\theta - \theta^\star) + \frac{KB}{2} = 0$$

$$\iff u'(\theta^\star)(\theta - \theta^\star) = -\frac{KB}{2} - M\mathrm{Err}(C) - u(\theta^\star)$$

$$\iff \theta - \theta^\star = \frac{-KB/2 - M\mathrm{Err}(C) - u(\theta^\star)}{u'(\theta^\star)}$$

$$\iff \theta = \theta^\star + \frac{-KB/2 - M\mathrm{Err}(C) - u(\theta^\star)}{u'(\theta^\star)}.$$

Then:

$$\mathrm{leb}\, S = \left|\left(\theta^\star + \frac{KB/2 + M\mathrm{Err}(C) - \ell(\theta^\star)}{\ell'(\theta^\star)}\right) - \left(\theta^\star + \frac{-KB/2 - M\mathrm{Err}(C) - u(\theta^\star)}{u'(\theta^\star)}\right)\right|$$

$$= \left|\theta^\star + \frac{KB/2 + M\mathrm{Err}(C) - \ell(\theta^\star)}{\ell'(\theta^\star)} - \theta^\star - \frac{-KB/2 - M\mathrm{Err}(C) - u(\theta^\star)}{u'(\theta^\star)}\right|$$

$$= \left|\frac{KB/2 + M\mathrm{Err}(C) - \ell(\theta^\star)}{\ell'(\theta^\star)} - \frac{-KB/2 - M\mathrm{Err}(C) - u(\theta^\star)}{u'(\theta^\star)}\right|$$

$$= \left|\left(\frac{u(\theta^\star)}{u'(\theta^\star)} - \frac{\ell(\theta^\star)}{\ell'(\theta^\star)}\right) + \left(\frac{KB/2 + M\mathrm{Err}(C)}{\ell'(\theta^\star)} + \frac{KB/2 + M\mathrm{Err}(C)}{u'(\theta^\star)}\right)\right|$$

$$\leq \left|\frac{u(\theta^\star)}{u'(\theta^\star)} - \frac{\ell(\theta^\star)}{\ell'(\theta^\star)}\right| + \left|\frac{KB/2 + M\mathrm{Err}(C)}{\ell'(\theta^\star)} + \frac{KB/2 + M\mathrm{Err}(C)}{u'(\theta^\star)}\right|$$

$$= \left|\frac{u(\theta^\star)}{u'(\theta^\star)} - \frac{\ell(\theta^\star)}{\ell'(\theta^\star)}\right| + \left|\left(\frac{1}{\ell'(\theta^\star)} + \frac{1}{u'(\theta^\star)}\right)(KB/2 + M\mathrm{Err}(C))\right|$$

$$= \left|\frac{u(\theta^\star)}{u'(\theta^\star)} - \frac{\ell(\theta^\star)}{\ell'(\theta^\star)}\right| + \left|\frac{1}{\ell'(\theta^\star)} + \frac{1}{u'(\theta^\star)}\right|(KB/2 + M\mathrm{Err}(C)).$$

Finally, by adding and subtracting the boundaries of the interval in expectation:

$$\left|\frac{u(\theta^\star)}{u'(\theta^\star)} - \frac{\ell(\theta^\star)}{\ell'(\theta^\star)}\right| + \left|\frac{1}{\ell'(\theta^\star)} + \frac{1}{u'(\theta^\star)}\right|(KB/2 + M\mathrm{Err}(C))$$

$$= \left|\frac{\mathbb{E}[\sup\psi(C(X);\theta^\star)]}{u'(\theta^\star)} + \frac{u(\theta^\star) - \mathbb{E}[\sup\psi(C(X);\theta^\star)]}{u'(\theta^\star)} - \frac{\mathbb{E}[\inf\psi(C(X);\theta^\star)]}{\ell'(\theta^\star)} - \frac{\ell(\theta^\star) - \mathbb{E}[\inf\psi(C(X);\theta^\star)]}{\ell'(\theta^\star)}\right|$$

$$+ \left|\frac{1}{\ell'(\theta^\star)} + \frac{1}{u'(\theta^\star)}\right|(KB/2 + M\mathrm{Err}(C))$$

$$= \left|\frac{\mathbb{E}[\sup\psi(C(X);\theta^\star)]}{u'(\theta^\star)} - \frac{\mathbb{E}[\inf\psi(C(X);\theta^\star)]}{\ell'(\theta^\star)} + \frac{u(\theta^\star) - \mathbb{E}[\sup\psi(C(X);\theta^\star)]}{u'(\theta^\star)} - \frac{\ell(\theta^\star) - \mathbb{E}[\inf\psi(C(X);\theta^\star)]}{\ell'(\theta^\star)}\right|$$

$$+ \left|\frac{1}{\ell'(\theta^\star)} + \frac{1}{u'(\theta^\star)}\right|(KB/2 + M\mathrm{Err}(C))$$

$$\leq \left|\frac{\mathbb{E}[\sup\psi(C(X);\theta^\star)]}{u'(\theta^\star)} - \frac{\mathbb{E}[\inf\psi(C(X);\theta^\star)]}{\ell'(\theta^\star)}\right|$$

$$+ \left|\frac{u(\theta^\star) - \mathbb{E}[\sup\psi(C(X);\theta^\star)]}{u'(\theta^\star)} - \frac{\ell(\theta^\star) - \mathbb{E}[\inf\psi(C(X);\theta^\star)]}{\ell'(\theta^\star)}\right|$$

$$+ \left|\frac{1}{\ell'(\theta^\star)} + \frac{1}{u'(\theta^\star)}\right|(KB/2 + M\mathrm{Err}(C)).$$

$$\leq \left|\frac{\mathbb{E}[\sup\psi(C(X);\theta^\star)]}{u'(\theta^\star)} - \frac{\mathbb{E}[\inf\psi(C(X);\theta^\star)]}{\ell'(\theta^\star)}\right|$$

$$+ \frac{|u(\theta^\star) - \mathbb{E}[\sup\psi(C(X);\theta^\star)]|}{|u'(\theta^\star)|} + \frac{|\mathbb{E}[\inf\psi(C(X);\theta^\star)] - \ell(\theta^\star)|}{|\ell'(\theta^\star)|}$$

$$+ \left|\frac{1}{\ell'(\theta^\star)} + \frac{1}{u'(\theta^\star)}\right|(KB/2 + M\mathrm{Err}(C)).$$

Now, since $G = \min\{|u'(\theta^\star)|, |\ell'(\theta^\star)|\}$, it follows that:

$$\left|\frac{1}{\ell'(\theta^\star)} + \frac{1}{u'(\theta^\star)}\right|(KB/2 + M\mathrm{Err}(C)) \leq \left(\frac{1}{|\ell'(\theta^\star)|} + \frac{1}{|u'(\theta^\star)|}\right)(KB/2 + M\mathrm{Err}(C))$$

$$\leq \frac{2}{G}(KB/2 + M\mathrm{Err}(C)) \leq \frac{1}{G}(KB + 2M\mathrm{Err}(C));$$

and

$$\frac{|u(\theta^\star) - \mathbb{E}[\sup \psi(C(X); \theta^\star)]|}{|u'(\theta^\star)|} + \frac{|\mathbb{E}[\inf \psi(C(X); \theta^\star)] - \ell(\theta^\star)|}{|\ell'(\theta^\star)|}$$

$$\leq \frac{|u(\theta^\star) - \mathbb{E}[\sup \psi(C(X); \theta^\star)]|}{G} + \frac{|\mathbb{E}[\inf \psi(C(X); \theta^\star)] - \ell(\theta^\star)|}{G}$$

$$= \frac{1}{G}\big(|u(\theta^\star) - \mathbb{E}[\sup \psi(C(X); \theta^\star)]| + |\mathbb{E}[\inf \psi(C(X); \theta^\star)] - \ell(\theta^\star)|\big).$$

Finally, we have to consider two cases:

(i) If $G = \min\{|u'(\theta^\star)|, |\ell'(\theta^\star)|\} = |u'(\theta^\star)|$, then

$$\left|\frac{\mathbb{E}[\sup \psi(C(X); \theta^\star)]}{u'(\theta^\star)} - \frac{\mathbb{E}[\inf \psi(C(X); \theta^\star)]}{\ell'(\theta^\star)}\right| = \frac{1}{G}\left|\mathbb{E}[\sup \psi(C(X); \theta^\star)] - \frac{u'(\theta^\star)}{\ell'(\theta^\star)}\mathbb{E}[\inf \psi(C(X); \theta^\star)]\right|$$

$$= \frac{1}{G}\left|\mathbb{E}[\sup \psi(C(X); \theta^\star)] - \mathbb{E}[\inf \psi(C(X); \theta^\star)] + \mathbb{E}[\inf \psi(C(X); \theta^\star)] - \frac{u'(\theta^\star)}{\ell'(\theta^\star)}\mathbb{E}[\inf \psi(C(X); \theta^\star)]\right|$$

$$\leq \frac{1}{G}\left|\mathbb{E}[\sup \psi(C(X); \theta^\star)] - \mathbb{E}[\inf \psi(C(X); \theta^\star)]\right| + \frac{1}{G}\left|\mathbb{E}[\inf \psi(C(X); \theta^\star)] - \frac{u'(\theta^\star)}{\ell'(\theta^\star)}\mathbb{E}[\inf \psi(C(X); \theta^\star)]\right|$$

$$= \frac{1}{G}\left|\mathbb{E}[\sup \psi(C(X); \theta^\star)] - \mathbb{E}[\inf \psi(C(X); \theta^\star)]\right| + \frac{1}{G}|\mathbb{E}[\inf \psi(C(X); \theta^\star)]|\left|1 - \frac{u'(\theta^\star)}{\ell'(\theta^\star)}\right|$$

$$\leq \frac{1}{G}\left|\mathbb{E}[\sup \psi(C(X); \theta^\star)] - \mathbb{E}[\inf \psi(C(X); \theta^\star)]\right| + \frac{1}{G}\left|1 - \frac{u'(\theta^\star)}{\ell'(\theta^\star)}\right|\max\{|a_{\theta^\star}|, |b_{\theta^\star}|\}$$

$$= \frac{1}{G}\left|\mathbb{E}[\text{leb hull}(\psi(C(X); \theta^\star))]\right| + \frac{1}{G}\left|1 - \frac{u'(\theta^\star)}{\ell'(\theta^\star)}\right|\max\{|a_{\theta^\star}|, |b_{\theta^\star}|\}$$

$$= \frac{1}{G}\mathbb{E}[\text{leb hull}(\psi(C(X); \theta^\star))] + \frac{1}{G}\left|1 - \frac{u'(\theta^\star)}{\ell'(\theta^\star)}\right|\max\{|a_{\theta^\star}|, |b_{\theta^\star}|\}$$

$$= \frac{1}{G}\mathbb{E}[\text{leb hull}(\psi(C(X); \theta^\star))] + \frac{1}{G}\left|1 - \frac{\min\{\ell'(\theta^\star), u'(\theta^\star)\}}{\max\{\ell'(\theta^\star), u'(\theta^\star)\}}\right|\max\{|a_{\theta^\star}|, |b_{\theta^\star}|\}.$$

(ii) If $G = \min\{|u'(\theta^\star)|, |\ell'(\theta^\star)|\} = |\ell'(\theta^\star)|$, then

$$\left|\frac{\mathbb{E}[\sup \psi(C(X); \theta^\star)]}{u'(\theta^\star)} - \frac{\mathbb{E}[\inf \psi(C(X); \theta^\star)]}{\ell'(\theta^\star)}\right| = \frac{1}{G}\left|\frac{\ell'(\theta^\star)}{u'(\theta^\star)}\mathbb{E}[\sup \psi(C(X); \theta^\star)] - \mathbb{E}[\inf \psi(C(X); \theta^\star)]\right|$$

$$= \frac{1}{G}\left|\mathbb{E}[\sup \psi(C(X); \theta^\star)] - \mathbb{E}[\inf \psi(C(X); \theta^\star)] + \frac{\ell'(\theta^\star)}{u'(\theta^\star)}\mathbb{E}[\sup \psi(C(X); \theta^\star)] - \mathbb{E}[\sup \psi(C(X); \theta^\star)]\right|$$

$$\leq \frac{1}{G}\left|\mathbb{E}[\sup \psi(C(X); \theta^\star)] - \mathbb{E}[\inf \psi(C(X); \theta^\star)]\right| + \frac{1}{G}\left|\frac{\ell'(\theta^\star)}{u'(\theta^\star)}\mathbb{E}[\sup \psi(C(X); \theta^\star)] - \mathbb{E}[\sup \psi(C(X); \theta^\star)]\right|$$

$$= \frac{1}{G}\left|\mathbb{E}[\sup \psi(C(X); \theta^\star)] - \mathbb{E}[\inf \psi(C(X); \theta^\star)]\right| + \frac{1}{G}|\mathbb{E}[\sup \psi(C(X); \theta^\star)]|\left|1 - \frac{\ell'(\theta^\star)}{u'(\theta^\star)}\right|$$

$$\leq \frac{1}{G}\left|\mathbb{E}[\sup \psi(C(X); \theta^\star)] - \mathbb{E}[\inf \psi(C(X); \theta^\star)]\right| + \frac{1}{G}\left|1 - \frac{\ell'(\theta^\star)}{u'(\theta^\star)}\right|\max\{|a_{\theta^\star}|, |b_{\theta^\star}|\}$$

$$= \frac{1}{G}\left|\mathbb{E}[\text{leb hull}(\psi(C(X); \theta^\star))]\right| + \frac{1}{G}\left|1 - \frac{\ell'(\theta^\star)}{u'(\theta^\star)}\right|\max\{|a_{\theta^\star}|, |b_{\theta^\star}|\}$$

$$= \frac{1}{G}\mathbb{E}[\text{leb hull}(\psi(C(X); \theta^\star))] + \frac{1}{G}\left|1 - \frac{\ell'(\theta^\star)}{u'(\theta^\star)}\right|\max\{|a_{\theta^\star}|, |b_{\theta^\star}|\}$$

$$= \frac{1}{G}\mathbb{E}[\text{leb hull}(\psi(C(X); \theta^\star))] + \frac{1}{G}\left|1 - \frac{\min\{\ell'(\theta^\star), u'(\theta^\star)\}}{\max\{\ell'(\theta^\star), u'(\theta^\star)\}}\right|\max\{|a_{\theta^\star}|, |b_{\theta^\star}|\}.$$

Combining everything, we get the desired bound. □

**Proposition A.6** (Proposition 2.7 in the main text). *For any $\alpha \in (0, 1)$, let $\widehat{C}_\alpha^{(\text{M}\ell)}$ be as in Equation 5 from the main text. Then $\widehat{C}_\alpha^{(\text{M}\ell)}$ is a valid confidence interval for $\theta^\star$, i.e.,*

$$\mathbb{P}\left[\theta^\star \in \widehat{C}_\alpha^{(\text{M}\ell)}\right] \geq 1 - \alpha.$$

*Proof.* Since the loss is differentiable, first note that it must hold that $\frac{\mathrm{d}}{\mathrm{d}\theta}\mathbb{E}[\ell(Y;\theta^\star)] = 0$. By the dominated convergence theorem we can exchange the expectation and derivative, and so

$$\frac{\mathrm{d}}{\mathrm{d}\theta}\mathbb{E}[\ell(Y;\theta^\star)] = \mathbb{E}\left[\frac{\mathrm{d}}{\mathrm{d}\theta}\ell(Y;\theta^\star)\right] = 0.$$

Note that now our set $\widehat{C}_\alpha^{(\text{M}\ell)}$ for M-estimation corresponds to a set $\widehat{C}_\alpha^{(\text{Z}\psi)}$ for Z-estimation, for $\psi(Y;\theta) := \frac{\mathrm{d}}{\mathrm{d}\theta}\ell(Y;\theta^\star)$. So by applying Proposition 2.5 (Proposition 2.5 in the main text), we conclude. $\square$

**Proposition A.7** (Power analysis for M-estimation). *Consider $\Theta \subset \mathbb{R}$ bounded by $B$ (i.e., for all $\theta, \theta' \in \Theta$, $\|\theta - \theta'\| \leq B$). Suppose that $\widehat{L}_{\theta,\alpha/2}^{(\text{M}\ell)}$ and $\widehat{U}_{\theta,\alpha/2}^{(\text{M}\ell)}$ are both $K$-smooth in $\theta$ (i.e., differentiable w.r.t. $\theta$, with $K$-Lipschitz derivative), $\widehat{L}_{\theta,\alpha/2}^{(\text{M}\ell)} \leq \widehat{U}_{\theta,\alpha/2}^{(\text{M}\ell)}$, $M_\theta \leq M$ for all $\theta$ and that $\frac{\mathrm{d}}{\mathrm{d}\theta}\widehat{L}_{\theta^\star,\alpha/2}^{(\text{M}\ell)}, \frac{\mathrm{d}}{\mathrm{d}\theta}\widehat{U}_{\theta^\star,\alpha/2}^{(\text{M}\ell)} \neq 0$. Then*

$$
\begin{aligned}
\mathrm{leb}\,\widehat{C}_\alpha^{(\text{M}\ell)} \leq \frac{1}{H_{\min}}\bigg( & \mathbb{E}[\mathrm{leb}\,\mathrm{hull}(\frac{\mathrm{d}}{\mathrm{d}\theta}\ell(C(X);\theta^\star))] + 2M\,\mathrm{Err}(C) \\
& + |\mathbb{E}[\inf \frac{\mathrm{d}}{\mathrm{d}\theta}\ell(C(X);\theta^\star)] - \widehat{L}_{\theta^\star,\alpha/2}^{(\text{M}\ell)}| + |\widehat{U}_{\theta^\star,\alpha/2}^{(\text{M}\ell)} - \mathbb{E}[\sup \frac{\mathrm{d}}{\mathrm{d}\theta}\ell(C(X);\theta^\star)]| \\
& + KB + \max\{a_{\theta^\star}, b_{\theta^\star}\}\,|1 - H_{\min}/H_{\max}|\bigg),
\end{aligned}
$$

*where $H_{\min} = \min\left\{|\frac{\mathrm{d}}{\mathrm{d}\theta}\widehat{L}_{\theta^\star,\alpha/2}^{(\text{M}\ell)}|, |\frac{\mathrm{d}}{\mathrm{d}\theta}\widehat{U}_{\theta^\star,\alpha/2}^{(\text{M}\ell)}|\right\}$ and $H_{\max} = \max\left\{|\frac{\mathrm{d}}{\mathrm{d}\theta}\widehat{L}_{\theta^\star,\alpha/2}^{(\text{M}\ell)}|, |\frac{\mathrm{d}}{\mathrm{d}\theta}\widehat{U}_{\theta^\star,\alpha/2}^{(\text{M}\ell)}|\right\}$.*

*Proof.* As in Proposition 2.7 (Proposition 2.7 in the main text), we can convert the M-estimation CI to a Z-estimation one. This result then follows by just applying Proposition 2.6 (Proposition 2.6 in the main text). $\square$

**Proposition A.8** (Proposition 2.8 in the main text). *If $(E_0, E_1, \ldots)$ is a test supermartingale for the null $H_0$, then so is the sequence of conformal prediction-powered e-values $(E_0^{\text{ppi}-(C)}, E_1^{\text{ppi}-(C)}, \ldots)$ defined in Equation 6 from the main text.*

*Proof.* The sequence is guaranteed to be nonnegative due to the bounds on $\eta_i$, and starts at $E_0^{\text{ppi}-(C)} = 1$ by definition. So all that remains is to show that it is a supermartingale. For any point in time $i$, it follows:

$$
\begin{aligned}
\mathbb{E}[E_i^{\text{ppi}-(C)}|\mathcal{F}_{i-1}] &= \mathbb{E}\left[E_{i-1}^{\text{ppi}-(C)} \cdot \mathrm{rescale}_{\eta_i}\left(\inf e_i(C_i(X_i)) - (b_i - a_i)\mathrm{Err}(C_i)\right)|\mathcal{F}_{t-1}\right] \\
&= E_{i-1}^{\text{ppi}-(C)} \cdot \mathbb{E}\left[\mathrm{rescale}_{\eta_i}\left(\inf e_i(C_i(X_i)) - (b_i - a_i)\mathrm{Err}(C_i)\right)|\mathcal{F}_{t-1}\right] \\
&= E_{i-1}^{\text{ppi}-(C)} \cdot \left(1 + \eta_i\left(\mathbb{E}\left[\inf e_i(C_i(X_i)) - (b_i - a_i)\mathrm{Err}(C_i)|\mathcal{F}_{t-1}\right] - 1\right)\right).
\end{aligned}
$$

Now, by Lemma 2.1 (Lemma 2.1 in the main text),

$$
\begin{aligned}
& E_{i-1}^{\text{ppi}-(C)} \cdot \left(1 + \eta_i\left(\mathbb{E}\left[\inf e_i(C_i(X_i)) - (b_i - a_i)\mathrm{Err}(C_i)|\mathcal{F}_{t-1}\right] - 1\right)\right) \\
& \leq E_{i-1}^{\text{ppi}-(C)} \cdot \left(1 + \eta_i\left(\mathbb{E}\left[e_i(Y_i)|\mathcal{F}_{t-1}\right] - 1\right)\right) \\
& \leq E_{i-1}^{\text{ppi}-(C)} \cdot \left(1 + \eta_i\left(1 - 1\right)\right) = E_{i-1}^{\text{ppi}-(C)},
\end{aligned}
$$

where the last step follows under the null since the original e-values form a test supermartingale. $\square$

**Proposition A.9** (Proposition 2.9 in the main text). *If $e_i(\cdot) \in [a_i, b_i]$ for every $i$, then there exists some constant $r > 0$ independent of $n$ for which*

$$\mathbb{E}\left[\frac{1}{n}\log E_n^{\mathrm{ppi}-(C)}\right] \geq \mathbb{E}\left[\frac{1}{n}\log E_n\right] - \frac{1}{n}\sum_{i=1}^{n}\mathbb{E}[\mathrm{leb\ hull}(\log e_i(C_i(X_i)))]$$

$$-\frac{r}{n}\sum_{i=1}^{n}\mathbb{E}\left[h_i(\eta_i)\mathrm{Err}(C_i)\right] - \frac{r}{n}\sum_{i=1}^{n}\mathbb{E}\big[|1-\eta_i|\,|\inf e_i(C_i(X_i))-1|\big],$$

*where $h_i(\eta_i) = \log\frac{b_i}{a_i} + \eta_i(b_i - a_i)$, which is increasing in $\eta_i$.*

*Proof.* Let $\tau_i := \mathrm{rescale}_{\eta_i}(a_i - (b_i - a_i)\,\mathrm{Err}(C_i))$. First, note that $\log$ is $\frac{1}{\mathrm{rescale}_{\eta_i}(a_i-(b_i-a_i)\,\mathrm{Err}(C_i))}$-Lipschitz in $[\mathrm{rescale}_{\eta_i}(a_i - (b_i - a_i)\,\mathrm{Err}(C_i)), \mathrm{rescale}_{\eta_i}(b_i - (b_i - a_i)\,\mathrm{Err}(C_i))]$, and thus:

$$\log\mathrm{rescale}_{\eta_i}\Big(\inf e_i(C_i(X_i)) - (b_i - a_i)\,\mathrm{Err}(C_i)\Big)$$

$$\geq \log\inf e_i(C_i(X_i)) - \frac{\left|\mathrm{rescale}_{\eta_i}\Big(\inf e_i(C_i(X_i)) - (b_i - a_i)\,\mathrm{Err}(C_i)\Big) - \inf e_i(C_i(X_i))\right|}{\mathrm{rescale}_{\eta_i}(a_i - (b_i - a_i)\,\mathrm{Err}(C_i))}$$

and, by adding and subtracting $\mathrm{rescale}_{\eta_i}\big(\inf e_i(C_i(X_i))\big)$ and then invoking the triangular inequality, we get

$$\log\inf e_i(C_i(X_i)) - \frac{\left|\mathrm{rescale}_{\eta_i}\Big(\inf e_i(C_i(X_i)) - (b_i - a_i)\,\mathrm{Err}(C_i)\Big) - \inf e_i(C_i(X_i))\right|}{\mathrm{rescale}_{\eta_i}(a_i - (b_i - a_i)\,\mathrm{Err}(C_i))}$$

$$\geq \log\inf e_i(C_i(X_i)) - \frac{\left|\mathrm{rescale}_{\eta_i}\Big(\inf e_i(C_i(X_i)) - (b_i - a_i)\,\mathrm{Err}(C_i)\Big) - \mathrm{rescale}_{\eta_i}\Big(\inf e_i(C_i(X_i))\Big)\right|}{\mathrm{rescale}_{\eta_i}(a_i - (b_i - a_i)\,\mathrm{Err}(C_i))}$$

$$-\frac{\left|\inf e_i(C_i(X_i)) - \mathrm{rescale}_{\eta_i}\Big(\inf e_i(C_i(X_i))\Big)\right|}{\mathrm{rescale}_{\eta_i}(a_i - (b_i - a_i)\,\mathrm{Err}(C_i))}$$

$$= \log\inf e_i(C_i(X_i)) - \frac{\left|\eta_i\Big(\big(\inf e_i(C_i(X_i)) - (b_i - a_i)\,\mathrm{Err}(C_i)\big) - \inf e_i(C_i(X_i))\Big)\right|}{\mathrm{rescale}_{\eta_i}(a_i - (b_i - a_i)\,\mathrm{Err}(C_i))}$$

$$-\frac{\left|\inf e_i(C_i(X_i)) - \mathrm{rescale}_{\eta_i}\Big(\inf e_i(C_i(X_i))\Big)\right|}{\mathrm{rescale}_{\eta_i}(a_i - (b_i - a_i)\,\mathrm{Err}(C_i))}$$

$$= \log\inf e_i(C_i(X_i)) - \frac{|\eta_i(b_i - a_i)\,\mathrm{Err}(C_i)| + \left|\inf e_i(C_i(X_i)) - \mathrm{rescale}_{\eta_i}\Big(\inf e_i(C_i(X_i))\Big)\right|}{\mathrm{rescale}_{\eta_i}(a_i - (b_i - a_i)\,\mathrm{Err}(C_i))}$$

$$= \log\inf e_i(C_i(X_i)) - \frac{\eta_i(b_i - a_i)\,\mathrm{Err}(C_i) + \left|\inf e_i(C_i(X_i)) - \mathrm{rescale}_{\eta_i}\Big(\inf e_i(C_i(X_i))\Big)\right|}{\mathrm{rescale}_{\eta_i}(a_i - (b_i - a_i)\,\mathrm{Err}(C_i))}$$

$$= \log\inf e_i(C_i(X_i)) - \frac{\eta_i(b_i - a_i)\,\mathrm{Err}(C_i) + \left|\inf e_i(C_i(X_i)) - 1 - \eta_i\Big(\inf e_i(C_i(X_i)) - 1\Big)\right|}{\mathrm{rescale}_{\eta_i}(a_i - (b_i - a_i)\,\mathrm{Err}(C_i))}$$

$$= \log\inf e_i(C_i(X_i)) - \frac{\eta_i(b_i - a_i)\,\mathrm{Err}(C_i) + \left|\Big(\inf e_i(C_i(X_i)) - 1\Big) - \eta_i\Big(\inf e_i(C_i(X_i)) - 1\Big)\right|}{\mathrm{rescale}_{\eta_i}(a_i - (b_i - a_i)\,\mathrm{Err}(C_i))}$$

$$= \log\inf e_i(C_i(X_i)) - \frac{\eta_i(b_i - a_i)\,\mathrm{Err}(C_i) + \left|(1 - \eta_i)\Big(\inf e_i(C_i(X_i)) - 1\Big)\right|}{\mathrm{rescale}_{\eta_i}(a_i - (b_i - a_i)\,\mathrm{Err}(C_i))}$$

$$= \log\inf e_i(C_i(X_i)) - \frac{\eta_i(b_i - a_i)\,\mathrm{Err}(C_i) + |1 - \eta_i|\,|\inf e_i(C_i(X_i)) - 1|}{\mathrm{rescale}_{\eta_i}(a_i - (b_i - a_i)\,\mathrm{Err}(C_i))}.$$

It thus follows:

$$\mathbb{E}\left[\frac{1}{n}\log E_n^{\text{ppi}-(C)}\right]$$

$$= \mathbb{E}\left[\frac{1}{n}\log\prod_{i=1}^{n}\text{rescale}_{\eta_i}\left(\inf e_i(C_i(X_i)) - (b_i - a_i)\text{Err}(C_i)\right)\right]$$

$$\geq \mathbb{E}\left[\frac{1}{n}\sum_{i=1}^{n}\left[\log\inf e_i(C_i(X_i)) - \frac{\eta_i(b_i - a_i)\text{Err}(C_i) + |1 - \eta_i|\,|\inf e_i(C_i(X_i)) - 1|}{\text{rescale}_{\eta_i}(a_i - (b_i - a_i)\text{Err}(C_i))}\right]\right]$$

$$= \mathbb{E}\left[\frac{1}{n}\sum_{i=1}^{n}\log\inf e_i(C_i(X_i)) - \frac{1}{n}\sum_{i=1}^{n}\frac{\eta_i(b_i - a_i)\text{Err}(C_i) + |1 - \eta_i|\,|\inf e_i(C_i(X_i)) - 1|}{\text{rescale}_{\eta_i}(a_i - (b_i - a_i)\text{Err}(C_i))}\right]$$

$$= \frac{1}{n}\sum_{i=1}^{n}\mathbb{E}\left[\log\inf e_i(C_i(X_i))\right] - \frac{1}{n}\sum_{i=1}^{n}\mathbb{E}\left[\frac{\eta_i(b_i - a_i)\text{Err}(C_i) + |1 - \eta_i|\,|\inf e_i(C_i(X_i)) - 1|}{\text{rescale}_{\eta_i}(a_i - (b_i - a_i)\text{Err}(C_i))}\right]$$

Now, note:

$$\mathbb{E}\left[\log\inf e_i(C(X_i))\right] = \mathbb{E}\left[\inf\log e_i(C(X_i))\right]$$
$$= \mathbb{E}\left[\sup\log e_i(C(X_i))\right] - \left(\mathbb{E}\left[\sup\log e_i(C(X_i))\right] - \mathbb{E}\left[\inf\log e_i(C(X_i))\right]\right)$$
$$= \mathbb{E}\left[\sup\log e_i(C(X_i))\right] - \mathbb{E}\left[\text{leb hull}(\log e_i(C(X_i)))\right],$$

and, by Lemma 2.1 (Lemma 2.1 in the main text),

$$\mathbb{E}\left[\sup\log e_i(C(X_i))\right] \geq \mathbb{E}\left[\log e_i(Y)\right] - \mathbb{E}[\log b_i - \log a_i]\,\text{Err}(C_i).$$

Therefore, putting it all together, we get

$$\mathbb{E}\left[\frac{1}{n}\log E_n^{\text{ppi}-(C)}\right]$$

$$\geq \frac{1}{n}\sum_{i=1}^{n}\mathbb{E}\left[\log\inf e_i(C_i(X_i))\right] - \frac{1}{n}\sum_{i=1}^{n}\mathbb{E}\left[\frac{\eta_i(b_i - a_i)\text{Err}(C_i) + |1 - \eta_i|\,|\inf e_i(C_i(X_i)) - 1|}{\text{rescale}_{\eta_i}(a_i - (b_i - a_i)\text{Err}(C_i))}\right]$$

$$\geq \frac{1}{n}\sum_{i=1}^{n}\left(\mathbb{E}\left[\log e_i(Y)\right] - \mathbb{E}[\log b_i - \log a_i]\,\text{Err}(C_i) - \mathbb{E}\left[\text{leb hull}(\log e_i(C_i(X_i)))\right]\right)$$

$$- \frac{1}{n}\sum_{i=1}^{n}\mathbb{E}\left[\frac{\eta_i(b_i - a_i)\text{Err}(C_i) + |1 - \eta_i|\,|\inf e_i(C_i(X_i)) - 1|}{\text{rescale}_{\eta_i}(a_i - (b_i - a_i)\text{Err}(C_i))}\right]$$

$$= \frac{1}{n} \sum_{i=1}^{n} \mathbb{E}\left[\log e_i(Y)\right] - \frac{1}{n} \sum_{i=1}^{n} \mathbb{E}[\log b_i - \log a_i] \operatorname{Err}(C_i) - \frac{1}{n} \sum_{i=1}^{n} \mathbb{E}\left[\operatorname{leb hull}(\log e_i(C_i(X_i)))\right]$$

$$- \frac{1}{n} \sum_{i=1}^{n} \mathbb{E}\left[\frac{\eta_i(b_i - a_i)\operatorname{Err}(C_i) + |1 - \eta_i| \, |\inf e_i(C_i(X_i)) - 1|}{\operatorname{rescale}_{\eta_i}(a_i - (b_i - a_i)\operatorname{Err}(C_i))}\right]$$

$$= \mathbb{E}\left[\frac{1}{n}\log E_n\right] - \frac{1}{n} \sum_{i=1}^{n} \mathbb{E}[\log b_i - \log a_i] \operatorname{Err}(C_i) - \frac{1}{n} \sum_{i=1}^{n} \mathbb{E}\left[\operatorname{leb hull}(\log e_i(C_i(X_i)))\right]$$

$$- \frac{1}{n} \sum_{i=1}^{n} \mathbb{E}\left[\frac{\eta_i(b_i - a_i)\operatorname{Err}(C_i) + |1 - \eta_i| \, |\inf e_i(C_i(X_i)) - 1|}{\operatorname{rescale}_{\eta_i}(a_i - (b_i - a_i)\operatorname{Err}(C_i))}\right]$$

$$= \mathbb{E}\left[\frac{1}{n}\log E_n\right] - \frac{1}{n} \sum_{i=1}^{n} \mathbb{E}\left[(\log b_i - \log a_i) \operatorname{Err}(C_i) + \operatorname{leb hull}(\log e_i(C_i(X_i)))\right.$$

$$\left. + \frac{\eta_i(b_i - a_i)\operatorname{Err}(C_i) + |1 - \eta_i| \, |\inf e_i(C_i(X_i)) - 1|}{\operatorname{rescale}_{\eta_i}(a_i - (b_i - a_i)\operatorname{Err}(C_i))}\right]$$

$$= \mathbb{E}\left[\frac{1}{n}\log E_n\right] - \frac{1}{n} \sum_{i=1}^{n} \mathbb{E}[\operatorname{leb hull}(\log e_i(C_i(X_i)))]$$

$$- \frac{1}{n} \sum_{i=1}^{n} \mathbb{E}\left[(\log b_i - \log a_i) \operatorname{Err}(C_i) + \frac{\eta_i(b_i - a_i)\operatorname{Err}(C_i) + |1 - \eta_i| \, |\inf e_i(C_i(X_i)) - 1|}{\operatorname{rescale}_{\eta_i}(a_i - (b_i - a_i)\operatorname{Err}(C_i))}\right]$$

$$= \mathbb{E}\left[\frac{1}{n}\log E_n\right] - \frac{1}{n} \sum_{i=1}^{n} \mathbb{E}[\operatorname{leb hull}(\log e_i(C_i(X_i)))]$$

$$- \frac{1}{n} \sum_{i=1}^{n} \mathbb{E}\left[\operatorname{Err}(C_i) \log \frac{b_i}{a_i} + \frac{\eta_i(b_i - a_i)\operatorname{Err}(C_i) + |1 - \eta_i| \, |\inf e_i(C_i(X_i)) - 1|}{\operatorname{rescale}_{\eta_i}(a_i - (b_i - a_i)\operatorname{Err}(C_i))}\right].$$

Now, let $r = \max\{(\operatorname{rescale}_{\eta_i}(a_i - (b_i - a_i)\operatorname{Err}(C_i)))^{-1}, 1\}$. Then:

$$\mathbb{E}\left[\frac{1}{n}\log E_n\right] - \frac{1}{n} \sum_{i=1}^{n} \mathbb{E}[\operatorname{leb hull}(\log e_i(C_i(X_i)))]$$

$$- \frac{1}{n} \sum_{i=1}^{n} \mathbb{E}\left[\operatorname{Err}(C_i) \log \frac{b_i}{a_i} + \frac{\eta_i(b_i - a_i)\operatorname{Err}(C_i) + |1 - \eta_i| \, |\inf e_i(C_i(X_i)) - 1|}{\operatorname{rescale}_{\eta_i}(a_i - (b_i - a_i)\operatorname{Err}(C_i))}\right]$$

$$= \mathbb{E}\left[\frac{1}{n}\log E_n\right] - \frac{1}{n} \sum_{i=1}^{n} \mathbb{E}[\operatorname{leb hull}(\log e_i(C_i(X_i)))]$$

$$- \frac{1}{n} \sum_{i=1}^{n} \mathbb{E}\left[\operatorname{Err}(C_i) \log \frac{b_i}{a_i} + r\left(\eta_i(b_i - a_i)\operatorname{Err}(C_i) + |1 - \eta_i| \, |\inf e_i(C_i(X_i)) - 1|\right)\right]$$

$$\geq \mathbb{E}\left[\frac{1}{n}\log E_n\right] - \frac{1}{n} \sum_{i=1}^{n} \mathbb{E}[\operatorname{leb hull}(\log e_i(C_i(X_i)))]$$

$$- \frac{1}{n} \sum_{i=1}^{n} \mathbb{E}\left[r\operatorname{Err}(C_i) \log \frac{b_i}{a_i} + r\left(\eta_i(b_i - a_i)\operatorname{Err}(C_i) + |1 - \eta_i| \, |\inf e_i(C_i(X_i)) - 1|\right)\right]$$

$$= \mathbb{E}\left[\frac{1}{n}\log E_n\right] - \frac{1}{n} \sum_{i=1}^{n} \mathbb{E}[\operatorname{leb hull}(\log e_i(C_i(X_i)))]$$

$$- \frac{r}{n} \sum_{i=1}^{n} \mathbb{E}\left[\operatorname{Err}(C_i) \log \frac{b_i}{a_i} + \eta_i(b_i - a_i)\operatorname{Err}(C_i) + |1 - \eta_i| \, |\inf e_i(C_i(X_i)) - 1|\right]$$

$$= \mathbb{E}\left[\frac{1}{n}\log E_n\right] - \frac{1}{n}\sum_{i=1}^{n}\mathbb{E}[\text{leb hull}(\log e_i(C_i(X_i)))]$$

$$- \frac{r}{n}\sum_{i=1}^{n}\mathbb{E}\left[\left(\log\frac{b_i}{a_i} + \eta_i(b_i - a_i)\right)\text{Err}(C_i) + |1 - \eta_i|\,|\inf e_i(C_i(X_i)) - 1|\right]$$

$$= \mathbb{E}\left[\frac{1}{n}\log E_n\right] - \frac{1}{n}\sum_{i=1}^{n}\mathbb{E}[\text{leb hull}(\log e_i(C_i(X_i)))]$$

$$- \frac{r}{n}\sum_{i=1}^{n}\mathbb{E}\left[h_i(\eta_i)\text{Err}(C_i) + |1 - \eta_i|\,|\inf e_i(C_i(X_i)) - 1|\right]$$

$$= \mathbb{E}\left[\frac{1}{n}\log E_n\right] - \frac{1}{n}\sum_{i=1}^{n}\mathbb{E}[\text{leb hull}(\log e_i(C_i(X_i)))]$$

$$- \frac{r}{n}\sum_{i=1}^{n}\mathbb{E}\left[h_i(\eta_i)\text{Err}(C_i)\right] - \frac{r}{n}\sum_{i=1}^{n}\mathbb{E}\big[|1 - \eta_i|\,|\inf e_i(C_i(X_i)) - 1|\big]. \qquad \square$$

# B  ADDITIONAL RESULTS

## B.1  ALGORITHMS ATOP E-VALUES

Beyond simple hypothesis testing, e-values can also be used as components of larger inference procedures. Notable examples include e-value-based confidence intervals/sequences, multiple testing procedures, as well as more involved examples such as change-point detection (Shin et al., 2022; Shekhar & Ramdas, 2023), test-time adaptation (Bar et al., 2024) and more. Generally speaking, by simply replacing the e-values in these predictions with our conformal prediction-powered e-values we obtain prediction-powered versions of our procedures, while retaining validity.

Formally, we a family of e-values $(E^{(\gamma)})_{\gamma\in\Gamma}$ indexed over $\Gamma$, and have an algorithm $\mathcal{A}((E^{(\gamma)})_{\gamma\in\Gamma})$ that operates atop this family. This algorithm comes endowed with some notion of validity, which should depend crucially on the validity of the underlying e-values:

**Assumption B.1.** If for all $\gamma \in \Gamma$, $E^{(\gamma)}$ is a valid e-value, then the algorithm $\mathcal{A}((E^{(\gamma)})_{\gamma\in\Gamma})$ is valid.

It then easily follows that, as long as the boundedness assumptions for the conformal prediction-powered e-values are satisfied, simply replacing the e-values with their conformal prediction-powered counterparts retains validity, while generally enhancing power:

**Proposition B.2.** *Suppose that for all $\gamma \in \Gamma$, $(E_0^{(\gamma)}, E_1^{(\gamma)}, \ldots)$ forms a test supermartingale. Then $\mathcal{A}((E^{\text{ppi}-(\gamma)})_{\gamma\in\Gamma})$ is valid.*

*Proof.* By Proposition 2.8 (Proposition 2.8 in the main text), for every $\gamma \in \Gamma$, $E^{\text{ppi}-(\gamma)}$ is a test supermartingale. Thus they are all valid e-values, making the procedure atop the conformal e-values valid. $\qquad\square$

We can also quantify the power of the procedure, but this generally requires us to consider the specifics of the algorithm over the e-values.

A special case worth highlighting is that of confidence sequences. We want to infer a parameter $\theta^\star \in \Theta$, and have a family of e-values $(E_n^{(\theta)})_{\theta\in\Theta}$. We then produce a confidence set via the following algorithm, for some significance level $\alpha$:

$$\mathcal{A}((E_n^{(\theta)})_{\theta\in\Theta}) := \left\{\theta \in \Theta : E_n^{(\theta)} \le 1/\alpha\right\}. \tag{7}$$

It then follows:

**Proposition B.3.** $\mathcal{A}((E_n^{\mathrm{ppi}-(\theta)})_{\theta \in \Theta})$ *is an anytime-valid confidence sequence for* $\theta^\star$. *I.e.,*

$$\mathbb{P}[\forall t, \; \theta^\star \in \mathcal{A}((E_n^{\mathrm{ppi}-(\theta)})_{\theta \in \Theta})] \geq 1 - \alpha.$$

*Proof.* Because each $E_n^{(\theta)}$ is valid, we get that each $E_n^{\mathrm{ppi}-(\theta)}$ is also valid. Then, using Ville's inequality:

$$\mathbb{P}[\forall t, \; \theta^\star \in \mathcal{A}((E_n^{\mathrm{ppi}-(\theta)})_{\theta \in \Theta})] = 1 - \mathbb{P}[\exists t \text{ such that } \theta^\star \notin \mathcal{A}((E_n^{\mathrm{ppi}-(\theta)})_{\theta \in \Theta})]$$
$$= 1 - \mathbb{P}[\exists t \text{ such that } E_n^{\mathrm{ppi}-(\theta^\star)} > 1/\alpha]$$
$$\geq 1 - \alpha. \qquad \square$$

## B.2 ESTIMATION IN HIGHER DIMENSIONS

We state here a multi-dimensional version of Lemma 2.1 (Lemma 2.1 in the main text). The remaining results follow analogously, as long as one uses multivariate confidence intervals where necessary.

Here, we take $\phi : \mathcal{Y} \to \mathbb{R}^d$, and let $\{e_1, \ldots, e_d\}$ be an orthonormal basis for $\mathbb{R}^d$ (e.g. the cannonical basis). Then:

**Lemma B.4.** *Let* $C : \mathcal{X} \to 2^{\mathcal{Y}}$ *be a set predictor and suppose that* $\langle \phi(Y), e_j \rangle \in [a_j, b_j]$ *for every* $j = 1, \ldots, d$ *almost surely; let* $M = \sum_{i=1}^d (b_j - a_j)$. *Then*

$$\mathbb{E}\left[\sum_{i=1}^d e_j \inf \langle \phi(C(X)), e_j \rangle\right] - M \operatorname{Err}(C) \leq \mathbb{E}[\phi(Y)] \leq \mathbb{E}\left[\sum_{i=1}^d e_j \sup \langle \phi(C(X)), e_j \rangle\right] + M \operatorname{Err}(C).$$

*Proof.* First, note that

$$\mathbb{E}[\phi(Y)] = \mathbb{E}\left[\sum_{j=1}^d e_j \langle \phi(Y), e_j \rangle\right] = \sum_{j=1}^d e_j \mathbb{E}\left[\langle \phi(Y), e_j \rangle\right]. \tag{8}$$

Now, for each $j = 1, \ldots, d$, by Lemma 2.1 (Lemma 2.1 in the main text),

$$\mathbb{E}\left[\inf \langle \phi(C(X)), e_j \rangle\right] - (b_j - a_j)\operatorname{Err}(C) \leq \mathbb{E}\left[\langle \phi(Y), e_j \rangle\right] \leq \mathbb{E}\left[\sup \langle \phi(C(X)), e_j \rangle\right] + (b_j - a_j)\operatorname{Err}(C);$$

plugging this back into Equation 8, we get the desired bounds. $\qquad \square$

## B.3 EMPIRICAL RESULTS ON THE IMPACT OF THE PREDICTIVE MODEL

For the purpose of conducting controlled experiments, we generate synthetic data from a simple statistical model following

$$Y = \beta^T X + \epsilon, \quad X \sim \mathcal{N}(0, 10 I_{5 \times 5}), \; \epsilon \sim \mathcal{N}(\mu, \sigma^2),$$

for fixed coefficients $\beta$ sampled from a $\mathcal{N}(0, I_{5 \times 5})$; for our predictive model, we use

$$\widehat{Y} = \beta^T X,$$

and conformalize with the absolute residual score.

This allows us to freely tweak the values of $\sigma^2$ (corresponding to exogenous noise) and $\mu$ (corresponding to bias of the predictive model). For all results below, we consider the task of inferring the median of $Y$ via Z-estimation.

Figure 4 shows the interval widths of our method and baselines over varying values of $\sigma^2$. We see that for relatively small amounts of exogenous noise we have results akin to those presented in Figure 1 in the main text; but, as the noise grows our method becomes less efficient, mainly due to the unavoidable growth of the conformal predictive sets.

In Figure 5 we see the interval widths of our method and baselines over varying choices of $\mu$. Again, for low levels bias (i.e., $\mu$ is close to zero) our findings are similar to that of Figure 1; but, as the bias increases our method degrades.

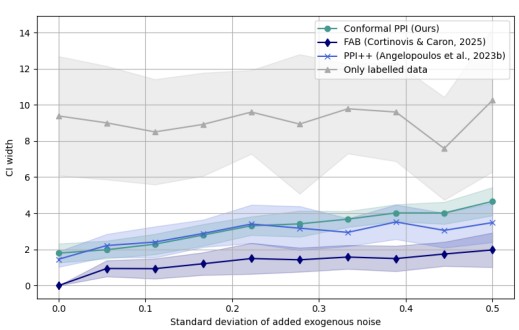

Figure 4: CI widths over varying levels of exogenous noise $\sigma^2$.

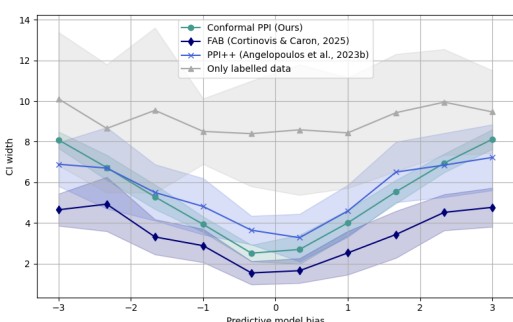

Figure 5: CI widths over varying levels of bias $\mu$.

## B.4 CHOICE OF $\gamma$

We analyze the sensitivity of confidence interval widths to the target miscoverage $\gamma$ using the data generating process described in Appendix B.3. As illustrated in Figure 6, the impact of $\gamma$ is intrinsically linked to the model's accuracy (governed here by the amount of exogenous noise, $\sigma$). For highly predictive models (low $\sigma$), decreasing $\gamma$ leads to a steady reduction in interval width, up until the point at which the conformal predictive sets degenerate (due to the calibration set size). Conversely, in high-noise regimes where the model lacks predictive power, the intervals become wide for low $\gamma$; in these cases, the trade-off shifts, and increasing $\gamma$ becomes advantageous. This empirical behavior aligns with the theoretical bounds established e.g. in Proposition 2.4.

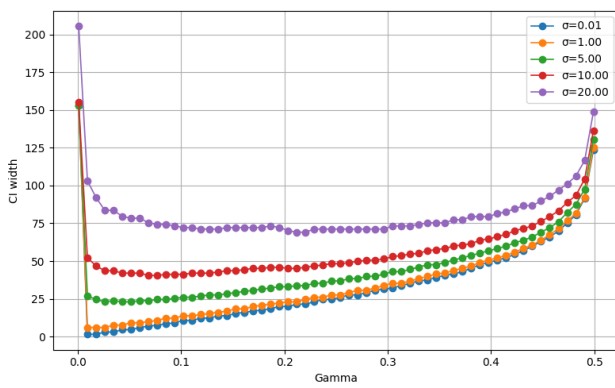

Figure 6: Sensitivity of CI widths to the choice of $\gamma$, across varying levels of exogenous noise $\sigma$.

### B.5 POWER AS A FUNCTION OF THE NUMBER SAMPLES

We provide here a result characterizing the width of our confidence intervals in terms of the number of unlabelled and labelled samples. This requires the choice of (i) a specific conformal calibration method; (ii) a method to produce the one-sided mean confidence intervals over the unlabelled samples. For tractability, we will also consider a specific well-specified predictive model: concretely, we assume that

$$Y = f(X) + \epsilon, \quad \text{for} \quad \epsilon \sim \text{Uniform}(-\delta, +\delta), \tag{9}$$

and take $f$ as our predictive model. This will allow us to precisely quantify the size of the conformal predictive sets. For the one-sided mean CIs, we will consider Hoeffding CIs due to their closed-form size formula.

We then have the following result:

**Proposition B.5.** *Under the data-generating process in Equation 9, using split conformal prediction with score $s(x, y) = |f(x) - y|$ and target miscoverage $\gamma \geq 1/(1 + n_{\text{cal}})$, and using our procedure described in Section 2.1 with $\phi(z) = z$, we have*

$$\mathbb{E}[\text{leb}\,\widehat{C}_\alpha^{(\mathbb{E}\phi)}] = 2\delta + 2(M - \delta)\gamma + 2M\sqrt{\frac{\log 2/\alpha}{2n_{\text{test}}}},$$

*where the expectation is with relation to both the calibration and test sets. Taking the optimal choice of $\gamma$ for this data generating process, we obtain*

$$\mathbb{E}[\text{leb}\,\widehat{C}_\alpha^{(\mathbb{E}\phi)}] = 2\delta + \frac{2(M - \delta)}{n_{\text{cal}} + 1} + 2M\sqrt{\frac{\log 2/\alpha}{2n_{\text{test}}}} = 2\delta + O(1/n_{\text{cal}}) + O(1/\sqrt{n_{\text{test}}}).$$

*Proof.* By Proposition 2.4,

$$\mathbb{E}[\text{leb}\,\widehat{C}_\alpha^{(\mathbb{E}\phi)}] = \mathbb{E}[\text{leb}\,\text{hull}(\phi(C(X)))] + 2M\gamma$$
$$+ (\mathbb{E}[\inf \phi(C(X))] - \mathbb{E}[\widehat{L}_{\alpha/2}^{(\mathbb{E}\phi)}]) + (\mathbb{E}[\widehat{U}_{\alpha/2}^{(\mathbb{E}\phi)}] - \mathbb{E}[\sup \phi(C(X))]).$$

Let us start by characterizing $C(X)$. Split conformal prediction with our score gives it the form

$$C(x) = \{y \in \mathcal{Y} : |f(x) - y| \leq t_\gamma\} = [f(x) - t_\gamma, f(x) + t_\gamma],$$

where

$$t_\gamma = \text{quantile}_{(1-\gamma)(1+n_{\text{cal}}^{-1})}(|f(X_1) - Y_1|, \ldots, |f(X_{n_{\text{cal}}}) - Y_{n_{\text{cal}}}|)$$
$$= \text{quantile}_{(1-\gamma)(1+n_{\text{cal}}^{-1})}(|\epsilon_1|, \ldots, |\epsilon_{n_{\text{cal}}}|),$$

assuming $(1 - \gamma)(1 + n_{\text{cal}}^{-1}) \leq 1$.

Now, since $\epsilon \sim \text{Uniform}(-\delta, +\delta)$, we have $|\epsilon|/\delta \sim \text{Uniform}(0, 1)$. Then the quantile corresponds to the $(1 - \gamma)(n_{\text{cal}} + 1)$-th order statistic, which for $|\epsilon|/\delta$ has distribution $\text{Beta}((1 - \gamma)(n_{\text{cal}} + 1), \gamma(n_{\text{cal}} + 1))$. So we have

$$\mathbb{E}[\text{leb}\,\text{hull}\,C(x)] = \mathbb{E}[2t_\gamma] = 2\delta\frac{(1 - \gamma)(n_{\text{cal}} + 1)}{(1 - \gamma)(n_{\text{cal}} + 1) + \gamma(n_{\text{cal}} + 1)} = 2\delta\frac{(1 - \gamma)(n_{\text{cal}} + 1)}{n_{\text{cal}} + 1} = 2\delta(1 - \gamma).$$

For the remaining terms, it follows:

$$\mathbb{E}[\inf \phi(C(X))] - \mathbb{E}[\widehat{L}_{\alpha/2}^{(\mathbb{E}\phi)}]$$

$$= \mathbb{E}[\inf \phi(C(X))] - \mathbb{E}\left[\frac{1}{n_{\text{test}}}\sum_{i=1}^{n_{\text{test}}}\inf \phi(C(X_i)) - M\sqrt{\frac{\log 2/\alpha}{2n_{\text{test}}}}\right] = M\sqrt{\frac{\log 2/\alpha}{2n_{\text{test}}}};$$

similarly, we obtain

$$\widehat{U}_{\alpha/2}^{(\mathbb{E}\phi)} - \mathbb{E}[\sup \phi(C(X))] = M\sqrt{\frac{\log 2/\alpha}{2n_{\text{test}}}}.$$

Putting everything together, we get

$$\mathbb{E}[\text{leb } \widehat{C}_\alpha^{(\mathbb{E}\phi)}] = 2\delta(1 - \gamma) + 2M\gamma + 2M\sqrt{\frac{\log 2/\alpha}{2n_{\text{test}}}}$$

$$= 2\delta + 2(M - \delta)\gamma + 2M\sqrt{\frac{\log 2/\alpha}{2n_{\text{test}}}}.$$

It must hold that $M \geq \delta$, so this is minimized for the lowest possible $\gamma$, given by $1/(n_{\text{cal}} + 1)$. This yields

$$\mathbb{E}[\text{leb } \widehat{C}_\alpha^{(\mathbb{E}\phi)}] = 2\delta + \frac{2(M - \delta)}{n_{\text{cal}} + 1} + 2M\sqrt{\frac{\log 2/\alpha}{2n_{\text{test}}}} = 2\delta + O(1/n_{\text{cal}}) + O(1/\sqrt{n_{\text{test}}}). \qquad \square$$

### B.6 INFERENCE OF UNBOUNDED MEANS

In Section 2.1 we outline a simple procedure for the prediction-powered inference of the mean of a bounded random variable. In this appendix, we'll show how we can leverage our procedure for e-values (Section 2.3) for the prediction-powered inference of the mean of an unbounded random variable. The key observation is that we can construct bounded e-values for the estimation of means from unbounded data with a test supermartingale structure, as we demonstrate below.

As with most e-value-based procedures, we will derive the method for testing a null hypothesis $H_0^{(\theta)} : \mathbb{E}[Y] = \theta$, but note that confidence intervals can be obtained by simply inverting the test (i.e., producing the CI $\{\theta \in \mathbb{R} : H_0^{(\theta)} \text{ is not rejected}\}$). Let $E_n$ be an e-value for $H_0^{(\theta)}$. There are many possible choices; for example, consider the Hoeffding-like e-value of (Waudby-Smith & Ramdas, 2020),

$$E_n := \prod_{i=1}^n \exp\left(\lambda_i(Y_i - \theta) - \frac{\lambda_i^2 \sigma^2}{2}\right) \quad \text{for some predictable sequence } \lambda_i \in \mathbb{R}; \qquad (10)$$

This is easily seen to be a valid test supermartingale for any $\sigma$-sub-Gaussian distribution:

**Proposition B.6.** *The random variable $E_n$ is a test supermartingale for $H_0^{(\theta)}$, for any $\sigma$-sub-Gaussian data distribution.*

*Proof.* Assume the null $H_0^{(\theta)}$, i.e., $\theta = \mathbb{E}[Y]$. Then $E_0 = 1$ by construction; so we just need to show that $E_n$ is a supermartingale. Indeed, at any step $n$,

$$\mathbb{E}[E_n \mid \mathcal{F}_{n-1}] = \mathbb{E}[E_{n-1} \cdot \exp(\lambda_n(Y_n - \theta) - \lambda_n^2 \sigma^2/2) \mid \mathcal{F}_{n-1}]$$

$$= E_{n-1} \cdot \mathbb{E}[\exp(\lambda_n(Y_n - \theta) - \lambda_n^2 \sigma^2/2) \mid \mathcal{F}_{n-1}];$$

Now, since the data is $\sigma$-sub-Gaussian, it holds (by definition) that $\mathbb{E}[\exp(\lambda(Y_n - \mathbb{E}[Y_n]))] \leq \exp(\lambda^2 \sigma^2/2)$ for any $\lambda \in \mathbb{R}$, and so

$$E_{n-1} \cdot \mathbb{E}[\exp(\lambda_n(Y_n - \theta) - \lambda_n^2 \sigma^2/2) \mid \mathcal{F}_{n-1}]$$

$$= E_{n-1} \cdot \mathbb{E}[\exp(\lambda_n(Y_n - \theta)) \mid \mathcal{F}_{n-1}] / \exp(\lambda_n^2 \sigma^2/2) \leq E_{n-1} \cdot 1 = E_{n-1}. \qquad \square$$

Sans sub-Gaussianity, one can appeal to more heavy-tailed assumptions (cf. e.g. (Waudby-Smith & Ramdas, 2020; Howard et al., 2018)), or appeal to central limit theory (e.g., Waudby-Smith et al. (2021)).

While $E_n$ is not itself bounded, we can truncate it at any $B > 0$ and rescale it about 1 without losing validity. To be precise:

**Proposition B.7.** *For any $B > 0$ and $0 > R > 1$, the process*

$$E_n := \prod_{i=1}^n \text{rescale}_R\left(\min\left\{\exp\left(\lambda_i(Y_i - \theta) - \frac{\lambda_i^2 \sigma^2}{2}\right), B\right\}\right), \quad \text{for some predictable sequence } \lambda_i \in \mathbb{R},$$

*with $\text{rescale}_R(e) = 1 + R \cdot (e - 1)$, is (i) a valid test supermartingale for $H_0^{(m)}$ for any $\sigma$-sub-Gaussian data distribution, and (ii) such that the components of the product over $i = 1, \ldots, n$ are all bounded in $[1 - R, 1 + R \cdot (B - 1)] \subset \mathbb{R}_{>0}$.*

*Proof.* To show that it is a valid test supermartingale: $E_0 = 1$ by construction. So again it suffices to show that $E_n$ is a supermartingale under the null. To this end, for any step $n$:

$$
\begin{aligned}
\mathbb{E}[E_n \mid \mathcal{F}_{n-1}] &= \mathbb{E}[E_{n-1} \cdot \mathrm{rescale}_R(\min\left\{\exp\left(\lambda_n(Y_n - \theta) - \lambda_n^2 \sigma^2/2\right), B\right\}) \mid \mathcal{F}_{n-1}] \\
&= E_{n-1} \cdot \mathbb{E}[\mathrm{rescale}_R(\min\left\{\exp\left(\lambda_n(Y_n - \theta) - \lambda_n^2 \sigma^2/2\right), B\right\}) \mid \mathcal{F}_{n-1}] \\
&= E_{n-1} \cdot \left(1 + R\left(\mathbb{E}[\min\left\{\exp\left(\lambda_n(Y_n - \theta) - \lambda_n^2 \sigma^2/2\right), B\right\} \mid \mathcal{F}_{n-1}] - 1\right)\right) \\
&\leq E_{n-1} \cdot \left(1 + R\left(\mathbb{E}[\exp\left(\lambda_n(Y_n - \theta) - \lambda_n^2 \sigma^2/2\right) \mid \mathcal{F}_{n-1}] - 1\right)\right) \\
&\leq E_{n-1} \cdot (1 + R(1 - 1)) = E_{n-1},
\end{aligned}
$$

where the last inequality follows as in Proposition B.6.

Boundedness follows immediately from simple computation: $\min\{\exp(\cdot), B\} \in [0, B]$ surely, and plugging this into $\mathrm{rescale}_R(\cdot)$ (which is increasing) gives the enunciated bounds. $\square$

With this, we have a valid test supermartingale for the null $H_0^{(\theta)}$ which is bounded, and thus our procedure in Section 2.3 can be directly applied.

## C  EXPERIMENT DETAILS

*Remark* C.1 (On solving for the CI bounds in Z- and M-estimation). For most Z-estimation problems (and M-estimation problems, once reduced to Z-estimation form) and one-sided mean CIs, the estimated bounds $\widehat{L}$ and $\widehat{U}$ on the influence function $\psi(y; \theta)$ are increasing in $\theta$. With this in mind, the inversion of the mean estimation bounds to produce our CIs can be done via standard bracketing and bisection procedures, guaranteeing correctness.

### C.1  PHISHING URL DATASET: MEAN ESTIMATION

**Dataset and split.** We employ the numeric subset of the Phishing URL corpus (Mohammad & McCluskey, 2012), containing $N = 235\,795$ labelled examples. The target parameter is the prevalence $\theta^\star = \mathbb{E}[Y]$ of phishing URLs. For every seed $s \in \{0, \dots, 99\}$ we create an independent **train/calibration/test** split as follows:

$$
\text{train} = 99.5\% \ (234\,616 \text{ samples}), \qquad \text{calibration} = 300, \qquad \text{test} = 879.
$$

The training labels are used solely to fit the predictive model; test labels are discarded.

**Predictive model.** An `XGBoost` classifier (default hyper-parameters, evaluation metric `logloss`) is trained on the numerical features of the training set:

```
model = xgb.XGBClassifier(eval_metric="logloss")
model.fit(X_tr, Y_tr)
```

**Conformity score.** Let $\hat{p}(x)$ be the model's predicted probability that $Y = 1$. For $(x, y) \in \mathcal{C}$ (calibration set) we use the conformity score

$$
s(x, y) = \begin{cases} \hat{p}(x), & y = 0, \\ 1 - \hat{p}(x), & y = 1. \end{cases}
$$

The miscoverage tolerance is $\mathrm{err} = 1.01/|\mathcal{C}|$. The $(1 - \mathrm{err})$-quantile of $\{s_i\}_{i \in \mathcal{C}} \cup \{+\infty\}$ yields the threshold $t$, from which we construct the prediction set $C(x) = \{0\}$ if $\hat{p}(x) \leq t$; $C(x) = \{1\}$ if $1 - \hat{p}(x) \leq t$; $C(x) = \{0, 1\}$ otherwise.

**Confidence-interval methods.** All intervals are built at significance level $\alpha = 0.01$ with a CLT-based constructor and target range $M = 1$.

For each seed we record the interval width with are reported in Figure 1(b). The full implementation is available at `supplementary/experiment1/mean_estimation.py`.

### C.2 GENE EXPRESSION DATASET: MEDIAN ESTIMATION

**Dataset and split.** In this experiments we focus on estimating the median of gene expression levels induced by yeast promoters sequences, we have access to labelled data and a transformer model from (Vaishnav et al., 2022), containing $N = 61\,150$ labelled examples. For every seed $s \in \{0, \dots, 99\}$ we create an independent **calibration/test** split as follows:

$$\text{calibration} = 10, \qquad \text{test} = 61140.$$

**Conformity score.** For $(x, y) \in \mathcal{C}$ (calibration set) we use the conformity score

$$s(x, y) = |y - f(x)|,$$

where $f(x)$ is the output of our pre-trained model.

The specified miscoverage level for conformal prediction is err $= 1.01/|\mathcal{C}|$. The $(1 - \text{err})$-quantile of $\{s_i\}_{i \in \mathcal{C}} \cup \{+\infty\}$ yields the threshold $t$, from which we construct the prediction set:

$$C(x) = (f(x) - t, f(x) + t).$$

**Confidence-interval methods.** All intervals are built at significance level $\alpha = 0.01$ with a CLT-based constructor and target range $M = 1$.

The full implementation is available at `supplementary/experiment1/quantile_estimation.py`.

### C.3 SECTION 3.2 IN THE MAIN TEXT

We use the dataset from (Borzooei & Tarokhian, 2023), which has 383 observations. We split 60% of these for statistical inference with our method; the remaining 40% are split into a training set (70%) and a testing set (30%). On the training set, we train an XGBoost model with default hyperparameters. On the test set, we calibrate a conformal predictor using the same conformity score we have used for classification, first with usual split conformal prediction and then with the differentially private conformal prediction method of (Angelopoulos et al., 2021). For the conformal calibrations, we use a target coverage of 2.5%.

The full implementation is available at `supplementary/experiment2/diff_priv.py`

### C.4 SECTION 3.3 IN THE MAIN TEXT

**Data, split and models** We use the dataset on forest cover type prediction of (Blackard, 1998). This dataset has $N = 581\,012$ samples. We then split 60% of the data for training and validating our model: 75% ($261\,455$) of that goes to training a Random Forest classifier and 25% ($87\,152$) to estimating a validation 0-1 loss. The remaining 40% ($232\,405$) of the data is used for our online risk monitoring (but only the first $100\,000$ of these are shown in the plot). Also on the validation set we train a residual model to predict the probability of whether the model made a correct prediction (i.e., predict the conditional 0-1 loss).

We setup two data streams: one unmodified, and another increasingly poisoned to simulate a harmful distribution shift. For this poisoning, at each point we flip a coin with probability $((t + 1)/5 + 0.1)^2 \mathbb{1}[t \geq 20\%]$, where $t \in [0, 1]$ indicates how far along in the experiment we are. If this coin falls heads (which can only happen after $t \geq 20\%$), then instead of using the real data we swap for a randomly chosen sample from a problematic set. This problematic set of samples is determined by those that our residual model predicts as at least 50% likely to be incorrect.

**Online conformal prediction** For conformal prediction, we use the same score as in the prior classification tasks, over our residual model. For the online conformal prediction method of (Angelopoulos et al., 2024) we use as hyperparameters $\epsilon = 0.3$ with an initial step size of 1.0, targeting a coverage of 0.1%.

**E-value & approximately log-optimal choice of the $\eta_i$s** Our base e-value is given by

$$e_i((X_i, Y_i)) := 1 + \lambda_i \left(\mathbb{1}[f(X_i) \neq Y_i] - (\text{ValRisk} + \epsilon_{\text{tol}})\right),$$

with $\lambda_i$ a predictable sequence of bets bounded in $(0, 1/(\text{ValRisk} + \epsilon_{\text{tol}}))$. When introducing the conformal prediction-powered modification, the overall e-values becomes

$$\prod_{i=1}^{n} \left(1 + \eta_i \left(\lambda_i \left(\mathbb{1}[f(X_i) \neq Y_i] - (\text{ValRisk} + \epsilon_{\text{tol}})\right) - (b_i - a_i)\text{Err}(C_i)\right)\right).$$

For the sake of simplicity, we take $\lambda_i = \eta_i$ at all steps. These $\eta_i$s are derived using an analogue of the aGRAPA criterion of (Waudby-Smith & Ramdas, 2020), meaning that we solve the first order optimality condition of the growth rate using a first-order Taylor approximation for $h(t) = 1/(1+t)$. The resulting $\eta_i$s are given by

$$\eta_i = \frac{\widehat{\mu}_i - (\text{ValRisk} + \epsilon_{\text{tol}}) - (b_i - a_i)\text{Err}(C_i)}{\widehat{\sigma}_i^2 + (\widehat{\mu}_i - (\text{ValRisk} + \epsilon_{\text{tol}}) - (b_i - a_i)\text{Err}(C_i))^2},$$

where $\widehat{\mu}$ and $\widehat{\sigma}^2$ are estimates of the mean and variance of the conformal imputations, respectively; we do these via exponentially weighted moving averages with $\alpha = 0.01$ in order to handle the non-i.i.d. structure.

The full implementation is available at `supplementary/experiment3/evalues.py`

