# OpenReview forum: "Extending Prediction-Powered Inference through Conformal Prediction"
_ICLR.cc/2026/Conference — Submitted to ICLR 2026_

### Official Review · Reviewer_RixV · 2025-10-30

**Soundness:** 2
**Presentation:** 2
**Contribution:** 2
**Rating:** 4
**Confidence:** 4

**Summary:**

This paper proposed a new Prediction-Powered Inference method to construct a confidence set for the unknown parameters. The new method is based on the conformal prediction set and relaxes the primary problem of constructing confidence bounds for the sup and inf quantities of the prediction set. Despite the finite-sample coverage guarantee, this approach could be very conservative if the range of the unknown parameter is wide.

**Strengths:**

This work connects the Prediction-Powered Inference and Conformal Prediction, two popular statistical frameworks for inference.

**Weaknesses:**

**1. The conservative issue of the proposed method.**

According to the confidence interval in Eq. (1), the width of the interval scales with the range of $\phi(Y)$, which could be very large in practical regression problems. To make this dependence vanish, Proposition 2.4 requires the coverage error $Err(C)$ of the conformal prediction set to tend to 0. However, for standard conformal prediction methods, if the coverage error converges to zero, the prediction set tends to be the whole label space. Considering the regression problem $Y = X + \epsilon$ with the prediction model $mu(X) = X$, if we use the absolute residual as the score, we get the CP set $C(X) = X \pm \hat{q}$, where $\hat{q}$ is the $(1-\alpha)(1+|\mathcal{C}|^{-1})$ quantile of calibration scores. To guarantee $Err(C) = 0$, we need $P(|Y - X| > \hat{q}) = P(|\epsilon| > \hat{q}) = 0$.

**2. The tradeoff on the miscoverage level of the prediction set is not well discussed in this paper.**

In addition to the limiting case after Proposition 2.4, the authors should discuss more about the tradeoff on the coverage level $\gamma$ of the conformal prediction set. In Proposition 2.4, $Err(C)$ is a decreasing function of $\gamma$, and other terms are increasing functions of $\gamma$. Are there any optimal choices in finite samples?

**3. About the use of labeled data.**

The labeled dataset in this paper is only used to construct the conformal prediction set. In the PPI paper (Angelopoulos et al., 2023a), the labeled dataset was directly used to build the confidence set of parameters. Hence, I'm wondering how we can leverage the labeled dataset to improve the confidence set. Also, the width characterization depending on the sample sizes of the labeled and unlabeled datasets should be added.

**4. About the experiments.**

In Appendix C.2, the range $M$ is set as $1$, is the target function bounded by $1$? In addition, all the specified miscoverage levels for conformal prediction are $1.01/|\mathcal{C}|$. What is the criterion to choose this level? The experiment results on different levels should be added. Also, there is no comparison with baseline methods in Figure 2. Overall, the comparison with existing PPI methods is not sufficient in experiments.

**Questions:**

See Weakness.

---

> ### Author Response · Authors · 2025-11-26
>
> We'd like to thank the reviewer for their remarks. Please refer to our comments below.
>
> **Conservativeness of the confidence intervals:** Indeed, our CIs can be a bit conservative. Nevertheless, our empirical results show that for sufficiently accurate models they are competitive with existing prediction-powered methods; we have added some experiments in this direction to Appendix B.3. Nevertheless, we emphasize that our method is the first and only one to naturally enable the construction of CIs with additional guarantees such as privacy, robustness to outliers, robustness to unknown distribution shifts, etc.
>
> **Choice of gamma:** It's not clear if there exists a "theoretically optimal" choice; we have, however, found the choice of $\gamma=1.01/n_{cal}$ (mentioned in the experiment details, Appendix C.1) to work reasonably well in practice (the 1.01 generally ensures that the conformal calibration does not degenerate to $+\infty$). Regarding the interplay between model quality and $\gamma$: for highly predictive models, smaller $\gamma$ values are viable and limited primarily by the calibration set size. Conversely, when the model lacks predictive power, the confidence sets tend to be large for small choices of $\gamma$, and so increasing $\gamma$ may be advantageous. We have included some empirical results in Appendix B.4 to illustrate these dynamics.
>
> **Use of labelled data:** Indeed, our method uses labelled data only to construct the conformal predictor, whereas vanilla PPI directly uses them in the construction of the confidence set. As shown in e.g. our Proposition 2.4, as long as the conformal predictive sets are sufficiently small, our approach is advantageous over not using the unlabelled data. We hope this answers your question, though otherwise please let us know. We have also added a new corollary in Appendix B.5 that characterizes the width of our CIs as a function of labelled and unlabelled sample sizes, as requested.
>
> **On the experiments:**
> - **On $M$:** Yes, in the phishing experiment we are inferring a probability, i.e., the mean of an indicator, and thus $M=1$ is justified. For the quantile problems, note that although the data is not bounded the pinball loss gradients are (cf. lines 221/222 on the new revision), and that is what we need for quantile inference with our method.
> - **On Figure 2:** The purpose of Figure 2 is to illustrate the use of our method with differential privacy (through the use of a differentially private conformal calibration method), quantifying how efficiency differs in comparison to the non-private version of our method. It does not make sense to compare with other PPI methods as none of them are differentially private (and it can be argued that making them DP would be a paper of its own). All of Section 3.1 (including Figure 1), along with the new results in Appendix B.3 are already dedicated to comparing with previous prediction-powered methods. All that said, if the reviewer judges it essential, we can add prior PPI methods as additional "non-private" baselines.

---

### Official Review · Reviewer_BRjj · 2025-10-30

**Soundness:** 2
**Presentation:** 3
**Contribution:** 3
**Rating:** 4
**Confidence:** 3

**Summary:**

The paper unifies prediction-powered inference with conformal prediction by replacing missing labels with a calibrated set predictor $C(x)$ and correcting with its miscoverage $\operatorname{Err}(C)=\operatorname{Pr}\{Y \notin C(X) \}$; the key inequality for any bounded $\varphi(Y) \in[a, b]$ (letting $M=b-a$ ) is $\mathbb{E}[\inf \varphi(C(X))]-M \operatorname{Err}(C) \leq \mathbb{E}[\varphi(Y)] \leq \mathbb{E}[\sup \varphi(C(X))]+M \operatorname{Err}(C)$, which yields a $(1- \alpha)$ confidence interval $\left[\widehat{L}_{\alpha / 2}-M \operatorname{Err}(C), \widehat{U}_{\alpha / 2}+M \operatorname{Err}(C)\right]$ and extends to Z-estimation via the confidence set $\left\{\theta: \widehat{L}_{\theta, \alpha / 2}-M_\theta \operatorname{Err}(C) \leq 0 \leq \widehat{U}_{\theta, \alpha / 2}+M_\theta \operatorname{Err}(C)\right\}$ with $M_\theta=b_\theta-a_\theta$; the same template lifts e-value procedures to prediction-powered, anytime-valid tests by inserting set-based  lower bounds into test supermartingales.
Overall, the method offers a single, general route to PPI with privacy, robustness, and distribution-shift guarantees, and provides the first general offline PPI with e-values-competitive where prior PPI works and enabling use cases it could not.

**Strengths:**

The paper presents a clear and well-motivated unification of prediction-powered inference and conformal prediction, and pushes this synthesis into realistic application regimes. The theoretical properties are carefully stated and developed. I particularly appreciate the authors’ positioning—and supporting results—that this is a general method for prediction-powered inference that comes with additional guarantees, and (to the best of my knowledge) the first use of conformal prediction for nonparametric statistical inference; moreover, the framework offers a principled route to deriving prediction-powered procedures with stronger guarantees such as privacy, robustness, and validity under continuous distribution shift.

**Weaknesses:**

1. The key observation in this paper is Lemma 2.1, which derives deterministic bounds for the target parameter, but these bounds depend on a bounded condition (the corresponding parameter M is usually unknown in applications). Although the truncation can be applied, the target parameter is implicitly changed after truncation. Moreover, it seems less possible to improve the proposed method along with the authors’ idea.
2. While principled, conformal prediction is known to be conservative in finite samples; in this paper the target is sandwiched between estimates of $\mathbb{E}[\inf \varphi(C(X))]$ and $\mathbb{E}[\sup \varphi(C(X))]$. In practice, constructing reliable one-sided bounds for these two expectations is itself more conservative than directly targeting $\mathbb{E}[\varphi(Y)]$, which further inflates the final interval length. The theory quantifies step-wise length inflation but does not yet mitigate intrinsic over-coverage or provide efficiency guarantees that materially narrow the intervals.
3. To address efficiency concerns, a focused simulation study should systematically contrast interval length and empirical coverage against PPI, PPI++, and FAB-PPI across controlled scenarios, to make the length coverage trade-off concrete.

**Questions:**

1. Choice of $\gamma$ and interval efficiency. Do you have guidance - either theoretical or empiricalfor choosing the conformal miscoverage target $\gamma$ so as to minimize the final interval width? Since the length scales with both the predictive-set diameter and $\operatorname{Err}(C)$, a discussion (or heuristic) balancing these two factors would be helpful. If feasible, could you report sensitivity curves of interval length versus $\gamma$ (and label budget), to illustrate the efficiency-coverage tradeoff?
2. Z-estimation inversion and solvability. In the Z-estimation setting, $\theta$ is obtained by inverting the one-sided bounds (yielding a confidence set). For canonical problems (e.g., mean/variance of a bounded outcome, logistic regression coefficient, quantile), can $\theta$ be solved in closed form or via a simple root-finder with guaranteed bracketing? A few worked examples (analytic or algorithmic) would clarify how practitioners should compute $\theta$ in common parametric tasks.
3. Comparisons under partial updates (Sec. 3.3). Even though Csillag et al. (2025) requires active data collection, it would still be informative to compare in settings where Prediction-Powered e-values are only updated when a new $Y$ is observed (i.e., no active querying). Such a study would help quantify ACI's conservatism and the practical gap between the two approaches under matched labelarrival processes.

---

> ### Author Response · Authors · 2025-11-26
>
> We'd like to thank the reviewer for their assessment; please refer to our comments below.
>
> **Boundedness:** We'd like to note that while the approach in our Section 2.1 (warmup) indeed fundamentally requires boundedness, we can still use our techniques to infer means of unbounded random variables by leveraging e-values. For example, consider the truncated and rescaled Hoeffding-like e-value for the mean of $\sigma$-sub-Gaussian data:
> $$ E_n := \prod_{i=1}^n \mathrm{rescale}\_R\left(\min\left\\{ \exp\left( \lambda\_i (Y\_i - \theta) - \frac{\lambda\_i^2 \sigma^2}{2} \right), B \right\\}\right), \quad \text{for some predictable sequence } \lambda\_i \in \mathbb{R}, B > 0 \text{ and } 0 < R < 1, $$
> for $\mathrm{rescale}_R(e) = 1 + R \cdot (e-1)$; this is a valid e-value with test supermartingale for any sub-Gaussian $Y_1, \ldots, Y_n$, as can be seen from Hoeffding's lemma / definition of sub-Gaussianity (cf. the new Appendix B.6). But, crucially, it is also such that the terms within the product are bounded as required in our method for e-values in Section 2.3 (even when the data $Y_1, \ldots, Y_n$ is unbounded), and so our technique can immediately be used. We have added a comment on this to Remark 2.2, along with a more detailed description in Appendix B.6.
>
> **Systematic comparisons:** Thank you for the suggestion; we have added in Appendix B.3 systematic comparisons on synthetic datasets varying the amount of exogenous noise and the accuracy of the predictive model. For small values of exogenous noise and predictive model bias our findings are in line with those in Figure 1 of the main text, but we see that as the noise becomes too large, or the bias becomes too great, our method starts performing worse than previous prediction-powered methods, but generally still better than not doing prediction-powered inference while enabling inference with additional guarantees.
>
> **Choice of $\gamma$:** It's not clear if there exists a "theoretically optimal" choice; we have, however, found the choice of $\gamma=1.01/n_{cal}$ (mentioned in the experiment details, Appendix C.1) to work reasonably well in practice (the 1.01 generally ensures that the conformal calibration does not degenerate to $+\infty$). Regarding the interplay between model quality and $\gamma$: for highly predictive models, smaller $\gamma$ values are viable and limited primarily by the calibration set size. Conversely, when the model lacks predictive power, the confidence sets tend to be large for small choices of $\gamma$, and so increasing $\gamma$ may be advantageous. We have included some empirical results in Appendix B.4 to illustrate these dynamics.
>
> **Z-estimation inversion and bracketing:** In the context of Z-estimation, suppose $\psi$ is nondecreasing (note that this corresponds to assuming convexity in the M-estimation setting, and is satisfied for all the usual instantiations of Z-estimation), and assume that the per-$\theta$ one-sided bounds are regular enough to maintain this monotonicity (this is the case for e.g. Hoeffding and CLT-based CIs). Then we are solving for the point at which this nondecreasing function crosses zero, which can be done with standard bracketing and bisection procedures. We have added a brief note on this to the experiment details appendix (Appendix C).
>
> **Comparisons with [Csillag et al., 2025] with partial updates:** It's not clear how such a comparison can be done. The method of [Csillag et al., 2025] requires knowledge of the label sampling probabilities (which they call $\pi_i(X_i)$), which they have access to due to their active data collection (flip a coin with probability $\pi_i(X_i)$ and collect the sample if it falls heads). This is fundamental to their approach and it is not clear how it can be used when it is not available. Does the reviewer have some particular experiment in mind?

---

### Official Review · Reviewer_qCJs · 2025-10-31

**Soundness:** 3
**Presentation:** 2
**Contribution:** 3
**Rating:** 6
**Confidence:** 3

**Summary:**

The paper proposes a unified framework that integrates prediction-powered inference with conformal prediction to achieve valid statistical inference. By performing imputation through conformal set-predictors, one can naturally inherit properties from the extensive literature on CP. The framework is developed for mean inference, Z- and M-estimation, and e-value-based inference. Extensive experiments have been done to asses the performance of the proposed method.

**Strengths:**

- Establishes a clear connection between prediction-powered inference and conformal prediction.
- The proposed framework is applicable to a wide range of inferential problems.
- Extensive real-data analyses have been conducted to demonstrate the effectiveness of the approach.

**Weaknesses:**

- A more in-depth discussion comparing the proposed method with existing prediction-powered inference and conformal prediction approaches would strengthen the paper. The current related work section mainly lists references without sufficient conceptual analysis.
- The statement following Proposition 2.3 could be clarified, as it is not immediately evident that the resulting confidence interval indeed inherits the properties of the set predictor.
- In Equation (1), it is unclear whether $Err(C)$ needs to be estimated. If so, the theoretical results should account for the corresponding estimation error.
- It would be valuable to discuss how the proposed method can be adapted to handle non-i.i.d. data and distributional shifts between labeled and unlabeled datasets, which are common in real-world applications.
- The paper could better highlight the connections between the presented theoretical results and the existing literature, emphasizing similarities, distinctions, and potential improvements.
- The boundedness assumption on $\psi(Y, \theta) $ is quite restrictive, as it excludes common distributions such as the normal distribution. Relaxing or justifying this assumption would improve the generality of the theoretical results.

**Questions:**

- It is suggested to move the proof sketch to the appendix to improve the readability and flow of the main text.
- The paper should clarify how to select $M\_\theta$ in practical applications.
- It would be helpful to discuss how the proposed method performs under different estimation models for $p(y∣x)$. Are the same predictive models used across all compared methods in the experiments?
- How the framework behaves under model misspecification, particularly when the predictive model is biased or underfitted?
- Additional explanations are needed on how the proposed procedure enables the use of a single private calibration for multiple inferences.

---

> ### Author Response · Authors · 2025-11-26
>
> We'd like to thank the reviewer for their comments.
>
> **How our method handles distribution shift, differential privacy, etc.:** We're happy to clarify this. Let us consider two concrete examples:
>
> 1. Privacy: here we have a conformal calibration method ensuring that the set-predictor $C(\cdot)$ is $(\epsilon,\delta)$-differentially private with relation to the calibration data. Then we want to show that our procedure inherits this, i.e., that our CI is differentially private with relation to the calibration data. Since our procedure interacts with the calibration data only through the already-private set-predictor $C(\cdot)$, this follows easily from the post-processing theorem of differential privacy, which establishes that the composition of a differentially-private procedure with any further computation retains privacy (cf. e.g. Proposition 2.1 in "The Algorithmic Foundations of Differential Privacy" by Dwork&Roth). Also note that since the labelled data is only used through the conformal predictor, changing the inference target (e.g. changing the $\phi$ in Section 2.1) requires no recalibration, justifying the use of a single private calibration for multiple inferences (which is unique to our procedure).
> 2. Distribution shift: consider now that there is a certain distribution shift between the labelled and unlabelled data, and that we use a conformal calibration method that handles this distribution shift (there are many options for this: e.g., https://arxiv.org/abs/1904.06019, https://arxiv.org/abs/2008.04267, https://arxiv.org/abs/2411.01596). Then $Err(C)$ will be controlled under the shifted distribution (thanks to the distribution shift-robust conformal calibration), and so our core lemma can be immediately applied; therefore, the resulting CI is valid even under the distribution shift.
>
> Note that both cases are immediate consequences of using an appropriate conformal calibration method along with the fundamental structure of the desired property; also note that this is easily adapted to essentially any form of robustness (to outliers, to fancy distribution shifts, adversarial attacks). We have added a footnote to the sentence following Proposition 2.3 clarifying the argument being made, thank you for your feedback.

---

> > ### Author Response · Authors · 2025-11-26
> >
> > **Biased/underfit predictive model:** In the main text, all experiments were run with the same model family; we agree that it would be interesting to have results exploring these aspects. To this end, we have now added some experiments analyzing the impact of exogenous noise and bias of the predictive model to Appendix B.3, which we hope assuages the reviewer's doubts. Overall, we find that for mild amounts of noise and bias, our model performs as illustrated in e.g. Figure 1 of the main text. When there are larger amounts of noise and/or larger bias (e.g., due to misspecification or undertraining) our procedure degrades in comparison to other prediction-powered methods, but often remains advantageous compared to not using the unlabelled data while still enabling prediction-powered inference with additional guarantees.
> >
> > **On Err(C):** For generality, our theoretical results assume knowledge of $Err(C)$, or at least an upper bound on it. This could be obtained either by estimation, or by leveraging the fact that $Err(C)$ is known to concentrate around $\gamma$ (see, e.g., the bounds in https://arxiv.org/abs/2205.03647). Either way, it is worth noting that generally speaking our method is reasonably robust to this choice (dubbed $\overline{Err}(C)$), since (following the proof sketch of the Lemma 2.1) we only need that
> > $$\mathbb{E}[\inf\phi(C(X))-\phi(Y)|Y\not\in C(X)]Err(C)\leq M\overline{Err}(C),$$
> > which holds iff
> > $$\mathbb{E}[\inf\phi(C(X))-\phi(Y)|Y\not\in C(X)]\leq M\cdot\frac{\overline{Err}(C)}{Err(C)}.$$
> > Since the expectation will typically be significantly less than $M$ (since $M$ is a worst-case bound for the variable inside the expectation), this will hold even when $\overline{Err}(C)$ is "misspecified" in the procedure.
> >
> > **Boundedness and $M_\theta$:** $M_\theta$ is typically quite simple to find: for example, if we are inferring a probability (i.e., mean of an indicator), $M=1$. Similarly, for $q$-quantile estimation via Z-estimation (cf. $\psi$ in line 221/222 on the new revision), we trivially have $\psi(Y; \theta) \in [-q, 1-q]$ for all $Y$ and $\theta$, so we can also take $M_\theta=1$.
> > Also, more broadly regarding means of unbounded random variables: we'd like to note that while the approach in our Section 2.1 (warmup) fundamentally requires boundedness, we can still use our techniques to infer means of unbounded random variables by leveraging e-values; we have added a comment on this to Remark 2.2, along with a more detailed discussion in Appendix B.6.
> >
> > **More discussion on related work:** Thank you for the suggestion. We will add a more in-depth discussion on previous methods for prediction-powered inference in a dedicated appendix, describing prior methods (especially vanilla PPI, PPI++ and FAB) and contrasting them to our solution. We hope this appropriately attends to the reviewer's concerns; if not, please let us know.

---

> > > ### Comment · Reviewer_qCJs · 2025-11-26
> > >
> > > Thanks to the authors for their efforts in addressing my concerns. Most of the issues have been resolved. However, I am maintaining my original score, as I am not fully convinced that the paper has reached the acceptance threshold.

---

> > > > ### Author Response · Authors · 2025-12-03
> > > >
> > > > We'd like to thank the reviewer for their prompt response; we are encouraged by the fact that our response has resolved most of their concerns, and are working to resolve any remaining doubts the reviewer may have had.
> > > >
> > > > To this end, we are preparing an in-depth appendix dedicated to presenting and discussing related works, spanning both prior prediction-powered inference methods (especially PPI, PPI++ and FAB) and conformal calibration methods & their guarantees, to be included in the camera-ready version. Furthermore, to further clarify the paragraph following Proposition 2.3, we will add an appendix instantiating our framework with various existing conformal prediction procedures, showing in detail how our procedure inherits/handles robustness, privacy, distribution shifts and non-i.i.d. data. We expect this to attend to the reviewer's remaining concerns, while rendering our paper more accessible to a broader audience.

---

### Official Review · Reviewer_Ax2N · 2025-10-31

**Soundness:** 3
**Presentation:** 2
**Contribution:** 3
**Rating:** 6
**Confidence:** 3

**Summary:**

The paper unifies conformal prediction with prediction-powered inference by imputing labels using calibrated conformal sets, enabling PPI to retain finite-sample validity while inheriting robustness, privacy, and distribution-shift tolerance. It instantiates the framework across means, Z/M-estimation, and e-values with competitive empirical performance, introduces the first general offline PPI scheme for e-values, and demonstrates high-impact applications in differentially private medical analysis and anytime-valid online risk monitoring.

**Strengths:**

The paper originally unifies prediction-powered inference with conformal prediction in a distribution-free manner using calibrated set predictors, transforming PPI into a general, modular framework that inherits robustness, privacy, and distribution-shift tolerance. It provides careful finite-sample guarantees for means, Z/M-estimation, and introduces the first general offline, anytime-valid PPI scheme for e-values, with power explicitly tied to set size and miscoverage. The exposition is clear and pedagogical, progressing from simple cases to general estimators. Experiments credibly validate the method, enabling differentially private medical analyses and anytime-valid risk monitoring without active data collection—both previously infeasible under standard PPI.

**Weaknesses:**

see detail in questions.

**Questions:**

1. In Propositions 2.3 (mean estimation) and 2.5 (Z-estimation), the method relies on the conformal predictor’s miscoverage rate $\operatorname{Err}(C) \approx \gamma$. If $\operatorname{Err}(C)$ significantly deviates from $\gamma$ (e.g., due to distribution shifts), how does the method ensure theoretical robustness of confidence intervals?

2. In the article, Appendix B.2 extends the framework to high-dimensional estimation tasks, but it only gives the multivariate version of the core lemma (Lemma B.4) without discussing computational efficiency. As the dimension increases, calculating the $\inf$ and $\sup$ of conformal prediction sets will face severe scalability issues. Are there optimization strategies to address this problem?

3. The proofs in the Appendix (e.g., Proposition A.4) rely on smoothness assumptions ($K$-Lipschitz derivatives). If $\psi(Y;\theta)$ is non-smooth (e.g., quantile loss), does the method become invalid? And is there any robustness guarantee?

4. In Section 2.1, the word ``disribution'' in ``Remark 2.2'' can be changed to ``distribution''.

---

> ### Author Response · Authors · 2025-11-26
>
> We'd like to thank the reviewer for their thorough review.
>
> **Deviations of Err(C):** To protect against distribution shifts, it suffices to use a conformal prediction calibration method that is robust to distribution shifts, e.g. https://arxiv.org/abs/2008.04267. This guarantees that $Err(C)$ will not deviate much.
>
> **Computational efficiency in high dimensions:** This is actually not so severe, and hinges mostly on the particular representation of the conformal predictive sets. In high dimensions, they would generally take the form of e.g. hypercubes or ellipsoids, both of which admit efficient solutions to the inf and sups. With this in mind, the computational complexity of applying the core lemma is just linear in the dimension.
>
> **$K$-Lipschitz derivatives assumption:** Proposition A.5, which bounds the interval sizes for Z-estimation, relies on $K$-smoothness, but Proposition A.4, which establishes validity of the procedure, does not. Hence, the procedure remains valid when these do not hold. In the bound for interval size this assumption is used to linearize the "inversion" of the bounds on the means, and could be replaced by variants such as assuming that the derivatives are $(\alpha, K)$-Hölder (rather than $K$-Lipschtiz; this is occasionally called $(\alpha+1,K)$-weak smoothness in the optimization literature); this would include, e.g., the quantile loss for $\alpha=0$.
>
> Thank you for pointing out the typo, we have fixed it in the new revision.

---

### Meta-Review · Area_Chair_3NYm · 2026-01-05

**Summary:**

Common concerns shared by the reviewers include: 1) the (theoretical) conservativeness of the method; 2) how the works under distribution shift or (adversarial) noise; 3) assumption of a known or estimated miscoverage rate for conformal prediction; 4) lack of in-depth discussion of related work.

Beyond these, for 2), the authors seem to rely purely on previous work on robust conformal prediction. However, it is worth noting that conformal methods under distribution shifts are not without their own assumptions, either on the type or the magnitude of shifts.

Therefore, I recommend a more careful investigation of how errors in conformal prediction or violation of the miscoverage upper bound influence the results, and a more careful discussion/investigation/experiment of the method's behavior under distribution shift or (adversarial) noise.

**Reviewer Concerns:**

The clarifications about the implementation details of the method itself are helpful. However, two concerns may remain: 1) the (theoretical) conservativeness of the method; 2) how the method works under distribution shift or adversarial noise. Both of these come from the fact that this is a "two step" method that the second step depends on the performance of the first step (or some assumptions on the first step). And the sensitivity to the errors in the first step or the violation of assumptions is a reasonable concern.

**Reviewer Scores:**

Reviewer qCJs explicitly mentioned that they will keep the score, as they are not sure the paper is ready for publication.

The other reviewers may likely keep there score, as the rebuttal is relatively brief and may oversimplified problems like distribution shifts.

---

### Decision · Program_Chairs · 2026-01-26

Reject